# Descent-Net: Learning Descent Directions for Constrained Optimization

## Abstract

Deep learning approaches, known for their ability to model complex relationships and fast execution, are increasingly being applied to solve large optimization problems. However, existing methods often face challenges in simultaneously ensuring feasibility and achieving an optimal objective value. To address this issue, we propose Descent-Net, a neural network designed to learn an effective descent direction from a feasible solution. By updating the solution along this learned direction, Descent-Net improves the objective value while preserving feasibility. Our method demonstrates strong performance on both synthetic optimization tasks and the real-world AC optimal power flow problem.

## 1 Introduction

Constrained optimization problems are ubiquitous in practical applications. While traditional optimization algorithms (Luenberger et al., 1984; Nocedal & Wright, 1999) offer strong theoretical guarantees, their computational efficiency often falls short when applied to modern large-scale problems. As a result, there is increasing interest in leveraging neural network-based methods to tackle constrained optimization tasks. In recent years, many emerging works have proposed end-to-end frameworks for solving constrained optimization problems, including Donti et al. (2017); Amos & Kolter (2017); Zhang & Ghanem (2018); Agrawal et al. (2019); Geng et al. (2020), etc.

This research direction falls under the broader framework of Learning to Optimize (L2O)(Bengio et al., 2021; Chen et al., 2022), which aims to leverage deep learning to improve the efficiency and scalability of optimization algorithms. Unlike traditional methods that rely on handcrafted update rules, L2O methods attempt to automatically learn optimization behaviors through data-driven approaches. However, most existing works consider unconstrained optimization problems. This motivates the development of more flexible frameworks that can incorporate feasibility into the learning dynamics while remaining scalable to large or structured problems.

In this work, we propose Descent-Net, a neural network architecture that takes as input the gradients of both the objective and constraint functions at a given feasible point. The network is trained to predict an effective descent direction and an appropriate step size, enabling objective improvement while maintaining feasibility. Initialized from feasible solutions obtained by methods such as DC3 (Donti et al., 2021), H-proj (Liang et al., 2024), etc., our method typically converges to a near-optimal solution in just a few update steps.

Our main contributions are summarized as follows:

- We design a new exact penalty subproblem that generates feasible descent directions for linearly constrained optimization problems, forming the foundation of our approach with theoretical convergence guarantees.

- We propose a neural network architecture, **Descent-Net**, which unrolls a projected subgradient method to solve the proposed subproblem. The network iteratively refines feasible solutions by learning effective descent directions at each step.

- We demonstrate the effectiveness of our approach through experiments on quadratic programs (QP) and a simple nonconvex variant of QP, both of which involve linear constraints. To further illustrate the applicability of Descent-Net beyond the linear setting, we also

evaluate it on the nonlinear AC optimal power flow problem. Across all experiments, our method consistently achieves solutions with relative errors on the order of $\mathbf{10^{-4}}$.

## 2 RELATED WORK

**Classical methods for constrained optimization.** Classical approaches to constrained optimization, including projected gradient descent, feasible direction methods (Zoutendijk, 1960; Topkis & Veinott, 1967), and primal-dual algorithms (Luenberger et al., 1984; Nocedal & Wright, 1999; Boyd et al., 2011), have been extensively studied and widely applied. These methods typically offer convergence guarantees under suitable assumptions, but often suffer from high iteration complexity and significant computational cost.

**Learning to optimize (L2O).** L2O seeks to replace hand-crafted optimization routines with learnable architectures that generalize across problem instances. Broadly speaking, L2O methods can be classified into model-free and model-based approaches (Chen et al., 2022). Model-free methods, such as those based on recurrent neural networks (e.g., LSTM) (Graves, 2014; Andrychowicz et al., 2016), aim to learn update rules directly from data. Model-based methods, on the other hand, incorporate algorithmic structure into the design of the network. Notable examples include LISTA (Chen et al., 2018b), unrolled manifold optimization algorithms (Gao et al., 2022). However, most existing L2O methods focus on unconstrained or simple constrained problems and fail to guarantee feasibility when applied to general constrained settings.

To address this, recent works incorporate constrained optimization structures into neural networks via projection layers (Yang et al., 2020; Liang et al., 2024) or differentiable optimization modules (Amos & Kolter, 2017; Agrawal et al., 2019; Bolte et al., 2021). However, these methods typically suffer from scalability and the need to solve nested optimization problems during training. Some approaches target special cases, such as linear constraints (Wang et al., 2023), but their applicability to more general problems remains limited. An alternative line of work draws inspiration from primal-dual methods, leading to neural architectures based on ADMM (Xie et al., 2019) and PDHG (Li et al., 2024). Such methods are usually evaluated by the KKT error, where feasibility and objective optimality are of the same order of magnitude, which makes them less suitable for scenarios requiring strict constraint satisfaction. Recent efforts have attempted to address this by designing networks that explicitly return feasible points (Donti et al., 2021; Wu et al., 2025); however, such methods still fall short of reaching near-optimal solutions in practice.

**Implicit layers.** A growing body of work explores the use of implicit neural architectures, including optimization layers (Amos & Kolter, 2017), neural ordinary differential equations (ODEs) (Chen et al., 2018a), and deep equilibrium models (DEQs) (Bai et al., 2019). These models define network outputs via the solution of fixed-point or optimization problems, allowing compact yet highly expressive representations. Despite their potential, these approaches often incur high computational costs during both forward and backward passes. In the context of constrained optimization, additional challenges arise when estimating gradients of projection operators, particularly in the presence of complex or nonconvex constraints. Approximate techniques such as gradient perturbation or stochastic sampling (Pogančić et al., 2019; Berthet et al., 2020) have been proposed, but typically come at the expense of increased variance and computational overhead.

## 3 PROBLEM SETUP

For any given data $x \in \mathbb{R}^d$, we solve the following constrained optimization problem

$$\min_{y \in \mathbb{R}^n} f_x(y), \quad \text{s.t.} \quad y \in \mathcal{C} := \{y \mid g_x(y) \leq 0, \ h_x(y) = 0\}, \tag{1}$$

where $f, g$, and $h$ are smooth (but not necessarily convex) functions that may depend on the input data $x$. We assume there are $m$ equality constraints and $l$ inequality constraints:

$$h_x(y) = [h_{x,1}(y), h_{x,2}(y), \ldots, h_{x,m}(y)]^T = 0,$$

$$g_x(y) = [g_{x,1}(y), g_{x,2}(y), \ldots, g_{x,l}(y)]^T \leq 0,$$

where $h_{x,i} : \mathbb{R}^n \to \mathbb{R}$ and $g_{x,j} : \mathbb{R}^n \to \mathbb{R}$ for all $i = 1, \ldots, m$ and $j = 1, \ldots, l$. We have the following common assumptions for this problem.

**Assumption 1.** *The feasible set $\mathcal{C}$ is non-empty and closed; the sub-level set $\{y \in \mathcal{C} \mid f_x(y) \le f_x(y_0)\}$ is bounded.*

**Assumption 2.** *We assume that at any feasible point $y$, the gradients of the equality constraints, $\nabla h_i(y)$, for $i = 1, 2, \ldots, m$, are linearly independent.*

We also assume that the Linear Independence Constraint Qualification (LICQ) holds, which guarantees that the Karush–Kuhn–Tucker (KKT) conditions are necessary for local optimality.

**Assumption 3** (LICQ). *Let $y^* \in \mathcal{C}$ be a local optimal point of problem (1). We assume that the set of active constraint gradients at $y^*$,*

$$\{\nabla h_i(y^*)\}_{i=1}^m \ \cup \ \{\nabla g_j(y^*)\}_{j \in \mathcal{A}(y^*)}, \quad \text{where } \mathcal{A}(y^*) := \{j \in \{1, \ldots, l\} \mid g_j(y^*) = 0\},$$

*is linearly independent.*

The notation $\mathcal{A}$ denotes the *active set*[1], i.e., the set of inequality constraints that are satisfied with equality.

**Assumption 4.** *Let $\mathcal{X} \subseteq \mathbb{R}^p$ be a compact set and assume that all training and test parameters satisfy $x \in \mathcal{X}$. For each $x \in \mathcal{X}$, consider the feasible set $\mathcal{C}$. We assume that:*

1. *(**Uniform boundedness of feasible sets**) There exists a compact set $Y \subseteq \mathbb{R}^n$ such that*
$$\mathcal{C} \subseteq Y \quad \text{for all } x \in \mathcal{X}.$$

2. *(**Smoothness and uniform gradient bound**) The functions $f_x, h_x, g_x$ are continuously differentiable in $y$, and the maps*
$$(x, y) \mapsto \nabla_y f_x(y) \quad \text{and} \quad (x, y) \mapsto \nabla_y g_x(y)$$
*are continuous on $\mathcal{X} \times Y$. Then, by compactness, there exist constants $L_f > 0$ and $L_g > 0$ such that*
$$\|\nabla_y f_x(y)\|_2 \le L_f \quad \text{and} \quad \|\nabla_y g_x(y)\|_2 \le L_g \quad \text{for all } x \in \mathcal{X}, \, y \in \mathcal{C}.$$

This assumption is reasonable. In practical training, the dataset is always finite, so there must exist a corresponding upper bound.

**Assumption 5.** *There exists a constant $\delta > 0$ such that for every $x \in \mathcal{X}$ and every feasible point $y \in \mathcal{C}$,*

$$\min_{j : g_{x,j}(y) < 0} (-g_{x,j}(y)) \ge \delta_g,$$

*with the convention that the minimum over an empty index set is $+\infty$ (i.e., when all inequality constraints are active).*

In fact, this assumption is not very strong. In practical calculations, we can set $\delta_g$ as an extremely small value (e.g. 1e-5) and consider a constraint as active when $0 \le -g_{x,j} < \delta_g$.

## 3.1 FEASIBLE DIRECTIONS METHOD

The method of feasible directions (MFD) was originally developed by Zoutendijk in the 1960s (Zoutendijk, 1960). However, a well-known drawback of MFD is that it may fail to converge due to the so-called jamming phenomenon. To address this issue, various fundamental modifications and extensions of MFD have since been proposed and studied (Zoutendijk, 1960; Topkis & Veinott, 1967; Pironneau & Polak, 1973; Luenberger et al., 1984). In this section, we briefly review the framework of MFD under the assumption that the constraints $h_x(y)$ and $g_x(y)$ are linear.

Based on the first-order approximation of the constraint functions, it can be inferred that, to maintain the feasibility of the solution, a suitable descent direction $d$ at the current iterate $y$ should satisfy the following conditions:

$$\begin{aligned}
\langle d, \nabla h_{x,i}(y) \rangle &= 0, \quad \text{for } i = 1, \ldots, m, \\
\langle d, \nabla g_{x,j}(y) \rangle &\le 0, \quad \text{for } j \in \mathcal{A} = \{1 \le j \le l : g_{x,j}(y) = 0\},
\end{aligned} \tag{2}$$

where $\nabla h_{x,i}$ denotes the gradient of the equality constraints, and $\nabla g_{x,j}$ corresponds to the inequality constraints.

---

[1]This is distinct from the standard definition of the active set, which typically includes the indices corresponding to the equality constraints.

**Zoutendijk Direction-Finding Subproblem(Zoutendijk, 1960)**  The Zoutendijk method computes a search direction $d \in \mathbb{R}^n$ at a feasible point $y$ by solving the following linear program:

$$\min_{d \in \mathbb{R}^n} \quad \nabla f_x(y)^\top d$$
$$\text{s.t.} \quad \nabla h_x(y)^\top d = 0, \quad \nabla g_{x,j}(y)^\top d \leq 0, \quad j \in \mathcal{A}, \quad \|d\|_\infty \leq 1. \tag{MFD}$$

Here, $\nabla h_x(y)^\top \in \mathbb{R}^{m \times n}$ denotes the Jacobian matrix of the equality constraints.

The first constraint ensures that the direction is tangent to the equality constraint, while the second maintains feasibility with respect to the active inequalities. The infinity norm constraint serves to normalize the direction and keep the subproblem bounded.

The step size is then chosen as the largest feasible value such that $y + \alpha d$ remains in the feasible set:

$$\bar{\alpha} = \max\{\alpha \in (0, 1] \mid y + \alpha d \in \mathcal{C}\}.$$

However, when the iterate approaches the boundary of the feasible region, the step size $\bar{\alpha}$ may become arbitrarily small, potentially causing convergence issues (Topkis & Veinott, 1967).

**Topkis–Veinott Uniformly Feasible Direction Subproblem (Topkis & Veinott, 1967)**  To resolve this issue, Topkis and Veinott proposed a uniformly feasible direction (UFD) formulation:

$$\min_{d \in \mathbb{R}^n} \quad \nabla f_x(y)^\top d$$
$$\text{s.t.} \quad \nabla h_x(y)^\top d = 0, \quad \nabla g_{x,j}(y)^\top d \leq -M \cdot g_{x,j}(y), \quad j = 1, \ldots, l,$$
$$\sum_{i=1}^n |d_i| = 1. \tag{UFD}$$

The main difference lies in the fact that all inequality constraints are considered, and a constant $M > 0$ is introduced. Notably, setting $M = \infty$ recovers the original formulation in (MFD). This modification ensures that the computed direction $d$ satisfies

$$g_{x,j}(y + \alpha d) \leq 0, \quad \text{for all } j, \quad \text{as long as } \alpha \leq \frac{1}{M},$$

thus providing a uniform lower bound on feasible step sizes and overcoming the stalling issues of the original method. It can be shown that the direction obtained from (UFD) is a feasible descent direction. Moreover, under the Assumption 3, any accumulation point of the iterates generated by this method (Zoutendijk, 1960; Faigle et al., 2013) satisfies the KKT conditions.

## 4 ALGORITHM

### 4.1 REFORMULATION OF UFD SUBPROBLEM

Our goal is to design a learning-to-optimize (L2O) algorithm for solving the structured problem described above. However, both the Zoutendijk and Topkis–Veinott methods require solving constrained subproblems at each iteration, which are not suitable for direct embedding into neural networks. To address this, we reformulate the subproblem by exact penalty method. This enables us to implement the solver as an unrolled optimization process of projected subgradient method, forming the basis of our L2O algorithm. In the following, we describe the reformulated subproblem and the corresponding L2O architecture.

Motivated by (MFD), we first formulate the following penalized subproblem:

$$\min_d \nabla f_x(y)^\top d + \sum_{j=1}^l c_j \max\left(\langle d, \nabla g_{x,j}(y)\rangle, -M g_{x,j}(y)\right), \quad \text{s.t.} \quad d \in \mathcal{D}, \tag{3}$$

where $\mathcal{D} = \{d : \|d\|_2 \leq 1 \text{ and } \langle d, \nabla h_{x,i}(y)\rangle = 0, \ \forall i = 1, \ldots, m\}$, $c_j > 0$ and $M > 0$ are the regularization parameters. The hinge penalty is exact if the parameter $c_j$ is large enough. We have the following result. The proof can be found in Appendix.

**Lemma 1** (Exact hinge penalty). *Given any feasible point $y \in \mathcal{C}$, denote $c_{\min} := \min_j c_j$. If we have*

$$c_{\min} > \frac{L_f}{M\delta_g}, \tag{4}$$

*where $c_{\min}$ is selected independently of $x$, then every global minimizer of (3) is optimal for the following $L_2$-norm UFD subproblem*

$$\min_{\|d\|_2 \le 1} \quad \nabla f_x(y)^\top d$$
$$\text{s.t.} \quad \nabla h_x(y)^\top d = 0, \quad \nabla g_{x,j}(y)^\top d \le -M \cdot g_{x,j}(y), \quad j = 1, \dots, l. \tag{UFD-L2}$$

A specific choice of $c_j$ that satisfies (4) is

$$c_j = \frac{\|\nabla f_x(y)\|_2}{\epsilon - \frac{1}{2}Mg_{x,j}(y)}, \quad j = 1, \dots, l. \tag{5}$$

The constant $\epsilon > 0$ is a small positive number added to ensure numerical stability during the calculations. Intuitively, when $g_{x,j}(y)$ is close to zero, which indicates that the point $y$ lies near the boundary of the $j$-th inequality constraint, the corresponding weight $c_j$ should be larger, as such constraints are more likely to be violated in subsequent updates. By assigning higher weights to these near-active constraints, the network is encouraged to prioritize directions $d$ that satisfy $\langle d, \nabla g_{x,j}(y)\rangle \le -Mg_{x,j}(y)$, which helps prevent constraint violations. We remark there are many possible surrogates for $c_j$, e.g., $c_j = \exp(-\delta g_{x,j}(y))$ or the softmax function.

**Proposition 1.** *Let $H \in \mathbb{R}^{n \times m}$ denote the matrix formed by the gradients of the equality constraints*

$$H = [\nabla h_{x,1}(y) \quad \cdots \quad \nabla h_{x,m}(y)].$$

*Then, under Assumption 2, the expression for the projection onto $\mathcal{D}$ is given by*

$$\mathcal{P}(d) = \begin{cases} \hat{d}, & \text{if } \|\hat{d}\|_2 \le 1, \\ \hat{d}/\|\hat{d}\|_2, & \text{otherwise,} \end{cases} \quad \text{where} \quad \hat{d} = d - H(H^\top H)^{-1}H^\top d. \tag{6}$$

Consequently, the procedure of the projected (sub)gradient method for solving this problem is as follows:

$$d_{k+1} = \mathcal{P}\big(d_k - \gamma_k \mathbf{u}_k\big), \tag{7}$$

where $\gamma_k > 0$ is the step size and $\mathcal{P}$ is the projection operator defined in (6). Let $\mathbf{1}_{\{\cdot\}}$ denotes the indicator function, the subgradient term $\mathbf{u}_k$ is given by

$$\mathbf{u}_k = \nabla f_x(y) + \sum_{j=1}^{l} c_j \mathbf{1}_{\{\langle d^k, \nabla g_{x,j}(y)\rangle \ge -Mg_{x,j}(y)\}} \nabla g_{x,j}(y). \tag{8}$$

Note that in many practical problems, the matrix $H$ is fixed across instances or does not change frequently. In such cases, the projection in (6) can be precomputed, making the computational cost manageable. Examples include decision-focused learning setups, where the equality constraints remain constant across instances (Tan et al., 2020). In other problems, the equality constraints are simple, allowing the projection to be computed efficiently; for instance, in classical portfolio optimization(Fabozzi et al., 2008), the budget constraint enables a straightforward projection.

## 4.2 DESIGN OF DESCENT MODULE

Directly solving problem (3) using the projected (sub)gradient method usually results in slow convergence, due to the use of diminishing step size. To address this, we propose Descent-Module, which is designed by unrolling the projected (sub)gradient algorithm. In our proposed network architecture, each layer takes the form of one iteration of the projected (sub)gradient method,

$$d_{k+1} = \mathcal{P}\big(d_k - \gamma_k T^k(\mathbf{u}_k)\big). \tag{9}$$

The key difference is that we apply learnable modules $T^k$ to the subgradient term $\mathbf{u}_k$, and the definition of $T^k$ is given as follows:

$$T^k(\mathbf{u}_k) = \mathbf{V}^k \text{ReLU}\big(\mathbf{W}^k \mathbf{u}_k + \mathbf{b}_1^k\big) + \mathbf{b}_2^k, \tag{10}$$

where $\mathbf{W}^k \in \mathbb{R}^{q \times n}$, $\mathbf{V}^k \in \mathbb{R}^{n \times q}$ and $\mathbf{b}_1^k \in \mathbb{R}^q$, $\mathbf{b}_2^k \in \mathbb{R}^n$ are the weight matrix and bias that we need to learn and $\text{ReLU}(x) = \max(x, 0)$. The design of the operator $T$ follows the work in Wu et al. (2024), where the authors theoretically demonstrate that such a network architecture possesses strong universal approximation capabilities.

The step size $\gamma_k$ is also set as a learnable parameter, which avoids the need for manual tuning, and we provide in Appendix A.8 the learned values of $\gamma_k$ across different layers in our experiments.

The architectures of the Descent Module are illustrated in Figure 1. Each Descent Module consists of $K$ Descent Layers sharing the same architecture and the input to the first layer is chosen as $d_0 = -\nabla f_x(y)$.

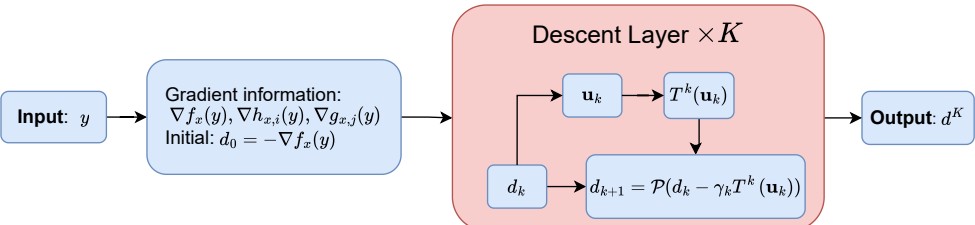

Figure 1: Overall structure of the Descent Module

We have the following theorem, and its proof is provided in the appendix.

**Theorem 4.1.** *Let $d^*$ be the optimal solution of Problem (3). For any $\varepsilon > 0$, there exists a $K_\varepsilon$-layer Descent-Module with a specific parameter assignment independent of $x$, whose output $d$ satisfies $|g(d) - g(d^*)| < \varepsilon$. Moreover, the number of layers satisfies $K_\varepsilon \leq \frac{C}{\varepsilon^2}$ for some constant $C > 0$.*

### 4.3 STEP SIZE

After obtaining the descent direction $d$ from the Descent module, we still need to determine a suitable step size. We assume that all constraints are linear. Since the Descent module contains the projection operator $\mathcal{P}$, the final descent direction $d$ produced by Descent module is orthogonal to the gradients of the equality constraints. Therefore, updating along $d$ will not violate the equality constraints.

We only need to ensure that the step size is not too large to violate the inequality constraints. For each linear inequality constraint $g_{x,j}$, we have:

$$g_{x,j}(y + \alpha d) = g_{x,j}(y) + \alpha \cdot \langle d, \nabla g_{x,j}(y) \rangle.$$

If $\langle d, \nabla g_{x,j}(y) \rangle > 0$, updating the solution along $d$ will increase $g_{x,j}$. To preserve the feasibility of the inequality constraint, i.e., $g_{x,j}(y + \alpha d) \leq 0$, the step size $\alpha$ must satisfy

$$\alpha \leq \frac{-g_{x,j}(y)}{\langle d, \nabla g_{x,j}(y) \rangle}.$$

Therefore, the maximum allowable step size is given by

$$\alpha_{\max} = \min_{j \in \mathcal{I}} \frac{-g_{x,j}(y)}{\langle d, \nabla g_{x,j}(y) \rangle}, \quad \text{where } \mathcal{I} = \{j \mid \langle d, \nabla g_{x,j}(y) \rangle > 0\}. \tag{11}$$

To guarantee a sufficient decrease of the objective value, the step size $\alpha$ should also satisfy $f_x(y + \alpha d) < f_x(y)$. To obtain a sufficient decrease in the objective value, we introduce a learnable parameter $\beta \in \mathbb{R}$ and use the sigmoid function $\sigma(\cdot)$ to map it into $(0, 1)$. We then use this factor to scale $\alpha_{\max}$, and the final update rule for $y$ is

$$y^{\text{new}} = y^{\text{old}} + \sigma(\beta)\alpha_{\max} \cdot d. \tag{12}$$

In addition, if the descent direction $d$ obtained from the Descent-Net is the optimal solution of Problem (3), Lemma 1 ensures that a fixed step size of $\alpha = 1/M$ is feasible. However, we found that such a fixed step size does not perform well in practice, and in the appendix A.6 we provide a comparison of different step-size selection strategies.

## 4.4 DESCENT-NET

Our Descent-Net consists of $S$ Descent Modules. The input of the network is an initial feasible solution $y_0$ of Problem (1). At each stage, the $s$-th module takes the gradient information at the current iterate $y_s$ and outputs a descent direction $d_s$, which is then used to update the solution to $y_{s+1}$ according to the update rule (12). By repeatedly updating the solution in this manner, the network finally produces a high-accuracy feasible solution $y_S$. The overall procedure of the proposed method is summarized in Algorithm 1.

---

**Algorithm 1** Descent-Net

---

1: **Input:** initial feasible point $y_0 \in \mathcal{C}$, $S$ modules and $K$ layers in each module.
2: **Learnable parameters:** $\Theta := \{\mathbf{V}^k, \mathbf{W}^k, \mathbf{b}_1^k, \mathbf{b}_2^k, \gamma_k\}_{k=0,1,\ldots,K-1}$ and $\{\beta_s\}_{s=0,\ldots,S-1}$.
3: **for** $s = 0, 1, \ldots, S-1$ **do**
4:      $d_0 = -\nabla f_x(y_s)$
5:      **for** $k = 0, \ldots, K-1$ **do**
6:          $\mathbf{u}_k = \nabla f_x(y_s) + \sum_{j=1}^l c_j \mathbf{1}_{\{\langle d_k, \nabla g_j(y_s)\rangle \geq -M g_{x,j}(y)\}} \nabla g_j(y_s)$
7:          $d_{k+1} = \mathcal{P}\left(d_k - \gamma_k T^k(\mathbf{u}_k)\right)$, where $\mathcal{P}$ is defined in (6) and $T^k$ is defined in (10) with
     parameters $\{\mathbf{V}^k, \mathbf{W}^k, \mathbf{b}_1^k, \mathbf{b}_2^k\}$
8:      **end for**
9:      $y_{s+1} = y_s + \sigma(\beta_s) \cdot \alpha_s d_K$ as defined by (12), where $\alpha_s$ is obtained by (11).
10: **end for**
11: Train the parameters with loss: $\ell_p(y) = f_x(y) + \lambda_g \|\text{ReLU}(g_x(y))\|_1 + \lambda_h \|h_x(y)\|_1$
12: **Output:** $y_S$.

---

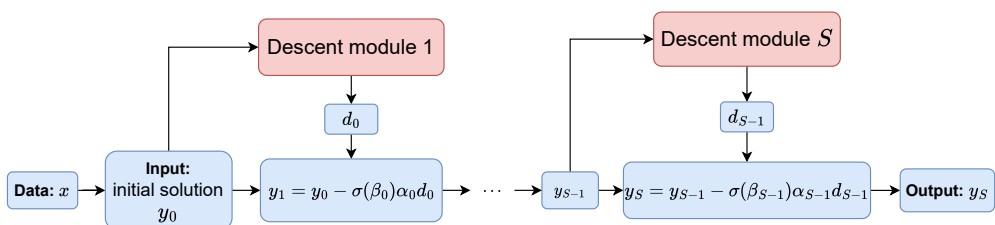

Figure 2: Architecture of the entire network.

**Theorem 4.2** (global convergence of the Descent-Net). *Suppose the Assumptions 1, 2, 3 hold. In addition, assume that $h_x, g_x$ are linear. Then there exists $K_\varepsilon$-layer Descent-Module with a specific parameter assignment independent of $x$ and $S > 0$ such that the Descent-Net generates a KKT conditions of the problem (1).*

The proof of the above theorem is given in Appendix. Although we assume linear constraints to establish the convergence guarantees, the proposed algorithm remains applicable in practice to problems with nonlinear constraints. Further improvements for handling general nonlinear constraints are left for future work.

## 5 EXPERIMENT

We evaluate our Descent-Net on three types of problems: convex quadratic programs, a simple class of non-convex optimization problems, and the AC optimal power flow (ACOPF) problem, with detailed experimental settings provided in Appendix A.2.

### 5.1 BASELINES AND EVALUATION CRITERIA

We compare our method against several benchmarks, including:

- **Optimizer**: Traditional numerical solvers, including `OSQP` (Stellato et al., 2020) and `qpth` (Amos & Kolter, 2017) for convex QPs, `IPOPT` (Wächter & Biegler, 2006) and `Knitro` (Byrd et al., 2006) for general nonlinear programs, and the `PYPOWER` solver (a Python port of `MATPOWER` (Zimmerman et al., 2005)) for ACOPF.
- **DC3**(Donti et al., 2021): The full DC3 framework that combines both completion and correction operators.
- **Projection method**:Trains an MLP and projects its output onto the feasible set. For problems with linear constraints (convex QP and simple non-convex cases), the projection is solved using OptNet (Amos & Kolter, 2017). For the ACOPF problem, the projection follows the differentiable solver of Chen et al. (2021).
- **Warm start**: The infeasible NN prediction is directly used as the warm-start for the optimizer of Chen et al. (2021), following the warm-starting schemes of Diehl (2019) and Baker (2019).
- **CBWF**(Wu et al., 2025): Inspired by the classical active set method, this approach explores the boundaries around inequality constraints and updates the initial solution to obtain a better objective value.

The performance of all methods is assessed according to the following criteria:

- **Feasibility:** measured by the average constraint violation of both equality and inequality constraints, i.e., $\frac{1}{m} \sum_{i=1}^{m} |h_{x,i}(y)|$ and $\frac{1}{l} \sum_{j=1}^{l} \text{ReLU}\left(g_{x,j}(y)\right)$.
- **Optimality:** measured by the average relative and absolute errors (in the $\ell_1$ norm) for both the solution and the objective value, where the optimal solution is approximated by optimizer.
- **Efficiency:** the computational time. It is worth noting that `OSQP`, `IPOPT`, `Knitro`, and `PYPOWER` only support sequential solving. For these solvers, we report the average runtime per instance to approximate full parallelization, while for other methods the runtime is measured with all test instances solved in parallel. For CBWF and Descent, the reported runtime includes both the time to obtain the initial solution and the time spent on refining the solution.

## 5.2 CONVEX QUADRATIC PROGRAMS

We first consider convex QPs with quadratic objectives and linear constraints:

$$\min_{y \in \mathbb{R}^n} \frac{1}{2} y^T Q y + p^T y, \quad \text{s.t. } Ay = x, \ Gy \leq h, \tag{13}$$

where $Q \in \mathbb{R}^{n \times n} \succeq 0$, $p \in \mathbb{R}^n$, $A \in \mathbb{R}^{n_{eq} \times n}$, $G \in \mathbb{R}^{n_{ineq} \times n}$, and $h \in \mathbb{R}^{n_{ineq}}$ are fixed. The input $x \in \mathbb{R}^{n_{eq}}$ varies across problem instances, and the goal is to approximate the optimal $y$ given $x$.

We generated 10,000 examples of $x$, and the experiment results are reported in Table 1. The initial solutions use those from DC3, and the final solutions produced by Descent-Net achieve a relative objective error of $2.6 \times 10^{-4}$. Moreover, Descent solves the instances about 58 times faster than the QP solver `qpth`. Note that the runtime reported for `OSQP` and `Knitro` corresponds to the average time per instance, as it only supports sequential solving, and is therefore less efficient than Descent-Net.

In addition, to further illustrate the effectiveness of Descent-Module, we examine the error between the descent direction $d$ and the optimal solution of its corresponding subproblem (3). The experimental results are provided in appendix A.7. We also compare Descent-Module with the original projected subgradient method, and the results are reported in appendix A.9. Furthermore, we include additional experiments on more QP instances in appendix A.4 and appendix A.5.

## 5.3 SIMPLE NON-CONVEX OPTIMIZATION

We now examine a simple non-convex adaptation of the quadratic program

$$\min_{y \in \mathbb{R}^n} \frac{1}{2} y^T Q y + p^T \sin(y), \quad \text{s.t. } Ay = x, \ Gy \leq h, \tag{14}$$

Table 1: Results on the convex QP task evaluated on the test set with 833 samples.

| Method | ineq. vio. | eq. vio. | sol. rel. err. | obj. rel. err. | Time (s) |
|---|---|---|---|---|---|
| Knitro | 0.0000 | 0.0000 | 0 | 0 | 0.0255 |
| OSQP | 0.0000 | 0.0000 | $7.9 \times 10^{-4}$ | $6.8 \times 10^{-6}$ | 0.0055 |
| qpth | 0.0000 | 0.0000 | $8.0 \times 10^{-4}$ | $6.8 \times 10^{-6}$ | 0.7540 |
| DC3 | 0.0000 | 0.0000 | $1.9 \times 10^{-1}$ | $1.1 \times 10^{-1}$ | 0.0038 |
| Projection method | 0.0000 | 0.0000 | $3.2 \times 10^{-2}$ | $8.4 \times 10^{-4}$ | 0.2124 |
| CBWF | 0.0000 | 0.0000 | $2.1 \times 10^{-1}$ | $6.6 \times 10^{-2}$ | 0.0366 |
| **DC3 + Descent (Ours)** | 0.0000 | 0.0000 | $1.2 \times 10^{-2}$ | $2.6 \times 10^{-4}$ | 0.0130 |

where $\sin(y)$ represents the component-wise application of the sine function to the vector $y$. Compared to problem (13), the only difference is that $y$ in the objective function is replaced with $\sin(y)$, which makes the problem non-convex.

The experimental results are presented in Table 2. The initial solutions use those from DC3, and the final solutions produced by Descent-Net achieve a relative objective error of $3.1 \times 10^{-4}$. Moreover, Descent-Net solves the instances approximately 10 times faster than the solver `IPOPT`.

Table 2: Results on the simple non-convex task evaluated on the test set with 833 samples.

| Method | ineq. vio. | eq. vio. | sol. rel. err. | obj. rel. err. | Time (s) |
|---|---|---|---|---|---|
| IPOPT | 0.0000 | 0.0000 | 0 | 0 | 0.1493 |
| DC3 | 0.0000 | 0.0000 | $2.2 \times 10^{-1}$ | $8.2 \times 10^{-2}$ | **0.0041** |
| Projection method | 0.0000 | 0.0000 | $5.4 \times 10^{-2}$ | $1.8 \times 10^{-3}$ | 0.2472 |
| CBWF | 0.0000 | 0.0000 | $2.6 \times 10^{-1}$ | $5.5 \times 10^{-2}$ | 0.0364 |
| **DC3 + Descent (Ours)** | 0.0000 | 0.0000 | $\mathbf{1.7 \times 10^{-2}}$ | $\mathbf{3.1 \times 10^{-4}}$ | 0.0144 |

## 5.4 ACOPF

The objective of the AC optimal power flow (AC-OPF) problem is to determine the optimal power generation that balances supply and demand while satisfying both physical laws and operational constraints of the network. A compact formulation of the AC-OPF problem is as follows:

$$\min_{p_g \in \mathbb{R}^n, \, q_g \in \mathbb{R}^n, \, v \in \mathbb{C}^n} \quad p_g^\top Q p_g + b^\top p_g$$

$$\text{s.t.} \quad p_g^{\min} \leq p_g \leq p_g^{\max}, \quad q_g^{\min} \leq q_g \leq q_g^{\max}, \quad v_m^{\min} \leq |v| \leq v_m^{\max}, \quad (15)$$
$$v_a^{\min} \leq \angle v_i - \angle v_j \leq v_a^{\max}, \quad |v_i(\overline{v_i} - \overline{v_j})\overline{w}_{ij}| \leq S_{ij}^{\max},$$
$$(p_g - p_d) + (q_g - q_d)i = \text{diag}(v)\overline{W}\,\overline{v}.$$

Here, $p_d, q_d \in \mathbb{R}^n$ denote the active and reactive power demands, and $p_g, q_g \in \mathbb{R}^n$ are the corresponding power generations. The complex bus voltage is represented by $v \in \mathbb{C}^n$. The nodal admittance matrix $W \in \mathbb{C}^{n \times n}$ encodes the network topology.

Since the equality constraints in this problem are nonlinear, a first-order approximation is not very accurate. As a result, even if the descent direction $d$ is orthogonal to the gradients of all equality constraints, the updated point may still fail to satisfy them. To address this issue, we adopt an equation completion approach, and the details are provided in the appendix A.10.

We conduct experiments on two ACOPF problem instances of different scales. Besides D-Proj (i.e., DC3), we use H-Proj (Liang et al., 2024) as another initialization strategy, and denote the corresponding solutions by $y^D$ and $y^H$.

D-Proj originally reduces violations of inequality constraints by performing a gradient descent step on the $\ell_2$ norm of constraint violations. In our experiments, we found that this gradient step is time-consuming and, in practice, often unnecessary. Therefore, we introduce an improved variant of

D-Proj by removing the gradient-descent step. This modification significantly reduces the computational time while maintaining comparable satisfaction of the inequality constraints. The optimized initialization obtained using this approach is denoted by $y^{D^*}$.

The results in Table 3 indicate that Descent-Net produces solutions with relative objective errors on the order of $10^{-4}$ across all cases. The relative error of the solution obtained by Descent-Net decreases only marginally compared to the initial point, which may be due to the non-convex nature of the ACOPF problem. Note that the runtime of PYPOWER is the average per instance since it solves sequentially, while Descent-Net solves instances in parallel, providing much higher efficiency.

Table 3: Results on the ACOPF task evaluated on the test set with 1024 samples.

| **30-bus system:** $n_{\text{eq}} = 60, n_{\text{ineq}} = 84$ | | | | | |
|---|---|---|---|---|---|
| Method | ineq. vio. | eq. vio. | sol. rel. err. | obj. rel. err. | Time (s) |
| PYPOWER | 0.0000 | 0.0000 | 0 | 0 | 0.5729 |
| Projection method | 0.0000 | 0.0000 | $5.6 \times 10^{-3}$ | $1.7 \times 10^{-2}$ | 0.0397 |
| Warm start | 0.0000 | 0.0000 | $5.5 \times 10^{-3}$ | $1.7 \times 10^{-2}$ | 0.0393 |
| D-Proj | 0.0000 | 0.0000 | $5.9 \times 10^{-3}$ | $1.9 \times 10^{-2}$ | 0.2442 |
| H-Proj | 0.0000 | 0.0000 | $5.8 \times 10^{-3}$ | $1.7 \times 10^{-2}$ | 0.2865 |
| **Descent** ($y_0 = y^D$) | 0.0000 | 0.0000 | $4.2 \times 10^{-3}$ | $3.6 \times 10^{-4}$ | 0.2619 |
| **Descent** ($y_0 = y^H$) | 0.0000 | 0.0000 | $3.5 \times 10^{-3}$ | $3.3 \times 10^{-4}$ | 0.3039 |
| **Descent** ($y_0 = y^{D^*}$) | 0.0000 | 0.0000 | $3.6 \times 10^{-3}$ | $2.8 \times 10^{-4}$ | 0.0434 |
| **118-bus system:** $n_{\text{eq}} = 236, n_{\text{ineq}} = 452$ | | | | | |
| Method | ineq. vio. | eq. vio. | sol. rel. err. | obj. rel. err. | Time (s) |
| PYPOWER | 0.0000 | 0.0000 | 0 | 0 | 1.2539 |
| Projection method | 0.0000 | 0.0000 | $1.5 \times 10^{-2}$ | $2.4 \times 10^{-3}$ | 0.3040 |
| Warm start | 0.0000 | 0.0000 | $9.3 \times 10^{-3}$ | $1.8 \times 10^{-3}$ | 0.3137 |
| D-Proj | 0.0000 | 0.0000 | $1.3 \times 10^{-2}$ | $2.4 \times 10^{-3}$ | 0.7542 |
| H-Proj | 0.0000 | 0.0000 | $1.4 \times 10^{-2}$ | $3.1 \times 10^{-3}$ | 0.6682 |
| **Descent** ($y_0 = y^D$) | 0.0000 | 0.0000 | $1.2 \times 10^{-2}$ | $2.5 \times 10^{-4}$ | 0.9480 |
| **Descent** ($y_0 = y^H$) | 0.0000 | 0.0000 | $1.4 \times 10^{-2}$ | $7.2 \times 10^{-4}$ | 0.8637 |
| **Descent** ($y_0 = y^{D^*}$) | 0.0000 | 0.0000 | $2.2 \times 10^{-3}$ | $3.0 \times 10^{-4}$ | 0.1622 |

## 6 FUTURE WORK

This work also points to several directions for further development. First, our theoretical results are established under the assumption of linear constraints. In the ACOPF experiments, the constraints are nonlinear, and although the method demonstrates strong empirical performance, extending the theoretical analysis to nonlinear or nonconvex settings represents an important direction for future work.

Second, to further demonstrate the advantages of our method, future work needs to validate it on problems of even larger scale than those considered here. However, obtaining feasible initial solutions for such instances is often challenging.

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

## A APPENDIX

### A.1 USE OF LARGE LANGUAGE MODELS

In preparing this manuscript, we used the large language model ChatGPT (GPT-5-mini) to assist with aspects of writing, including phrasing, grammar, and overall clarity of exposition. All scientific content, results, and interpretations are the original work of the authors. The use of ChatGPT was limited to writing assistance and did not influence the technical contributions or experimental results.

### A.2 EXPERIMENT SETTING

For the convex QPs and the simple non-convex problems, the parameters are generated as follows. The matrix $Q$ is diagonal with entries sampled i.i.d. from the uniform distribution on $[0, 1]$, while the entries of $A$ and $G$ are sampled i.i.d. from $N(0, 1)$. For each instance, the components of $x$ are drawn i.i.d. from the uniform distribution on $[-1, 1]$. To ensure that the generated problem has a feasible solution, we set $h_i = \sum_j |(GA^\dagger)_{ij}|$, where $A^\dagger$ denotes the Moore Penrose pseudoinverse of $A$. For the ACOPF experiments, we adopt the datasets provided in (Liang et al., 2024).

We summarize the hyperparameters used in our experiments in Table 4. Below we briefly describe several important parameters:

- $S$: the number of update steps performed in our Descent Net.
- $K$: determines the number of layers within each Descent module, controlling the expressive power of the network.
- $\lambda_h$: the penalty factor for equality constraint violations.
- $\lambda_g$: the penalty factor for inequality constraint violations.
- $q$: the hidden dimension of operator $T$, which specifies the capacity of feature transformation inside each descent step.
- $M, \epsilon$: parameters in $c_j$, which is defined in (5).

Table 4: Hyperparameters used in different experiments

| Hyperparameter | QP | Non-convex | ACOPF node=30 | ACOPF node=118 |
|---|---|---|---|---|
| Train size | 9167 | 9167 | 8976 | 18976 |
| Test size | 833 | 833 | 1024 | 1024 |
| Batch size | 64 | 64 | 512 | 512 |
| Epochs | 150 | 150 | 300 | 300 |
| Learning rate $lr$ | 0.001 | 0.001 | 0.01 | 0.01 |
| $S$ | 6 | 6 | 3 | 3 |
| $K$ | 3 | 3 | 3 | 3 |
| $\lambda_h$ | 5 | 5 | 5 | 5 |
| $\lambda_g$ | 5 | 5 | 5 | 5 |
| $q$ | 300 | 300 | 120 | 1080 |
| $M$ | 1 | 1 | 1 | 1 |
| $\epsilon$ | 0.0005 | 0.0005 | 0.0001 | 0.0001 |

For the parameters in the Descent module, we employ the Adam optimizer with an initial learning rate of $lr$, and reduce the learning rate by a factor of 0.1 at epochs 50, 100, and 150. The step-size adjustment parameter $\beta$ is updated separately using the SGD optimizer with a fixed learning rate of 0.01. In the ACOPF experiments, the gradient norm is clipped at a threshold of 1 to stabilize training, inspired by Zhang et al. (2019).

### A.3 EFFECT OF LAYER NUMBER $K$ AND DESCENT STEPS $S$

We conducted experiments on the convex quadratic program (13) to evaluate the performance of Descent-Modules with different numbers of layers $K$. The results are shown in Table 5. It can be

seen that increasing $K$ leads to a slight improvement in performance, but the gains are not significant. Considering computational efficiency, we ultimately choose $K = 3$ as the number of layers.

Table 5: Performance of Descent-Module with varying $K$

| Layer | ineq. vio. | eq. vio. | sol. rel. err. | obj. rel. err. |
|---|---|---|---|---|
| $K = 1$ | 0.0000 | 0.0000 | $9.8 \times 10^{-2}$ | $2.2 \times 10^{-2}$ |
| $K = 2$ | 0.0000 | 0.0000 | $9.3 \times 10^{-2}$ | $1.8 \times 10^{-2}$ |
| $K = 3$ | 0.0000 | 0.0000 | $9.2 \times 10^{-2}$ | $1.7 \times 10^{-2}$ |
| $K = 4$ | 0.0000 | 0.0000 | $8.9 \times 10^{-2}$ | $1.7 \times 10^{-2}$ |

With $K$ fixed at 3, we further examined the effect of different Descent steps $S$, as summarized in Table 6. When the number of update steps is 1, Descent-Net already achieves a solution with a relative error on the order of $10^{-2}$. Increasing the steps to 3 reduces the error to the $10^{-3}$ level, and further increasing to 6 reduces it to the $10^{-4}$ level.

Table 6: Performance of Descent-Net with varying $S$

| Descent Step | ineq. vio. | eq. vio. | sol. rel. err. | obj. rel. err. |
|---|---|---|---|---|
| $S = 1$ | 0.0000 | 0.0000 | $8.9 \times 10^{-2}$ | $1.7 \times 10^{-2}$ |
| $S = 2$ | 0.0000 | 0.0000 | $6.0 \times 10^{-2}$ | $1.1 \times 10^{-2}$ |
| $S = 3$ | 0.0000 | 0.0000 | $3.5 \times 10^{-2}$ | $3.4 \times 10^{-3}$ |
| $S = 4$ | 0.0000 | 0.0000 | $2.4 \times 10^{-2}$ | $1.6 \times 10^{-3}$ |
| $S = 5$ | 0.0000 | 0.0000 | $2.6 \times 10^{-2}$ | $2.0 \times 10^{-3}$ |
| $S = 6$ | 0.0000 | 0.0000 | $1.4 \times 10^{-2}$ | $4.5 \times 10^{-4}$ |

## A.4 ADDITIONAL EXPERIMENTS ON QUADRATIC PROGRAMS

We further evaluate the robustness and scalability of our method on more general quadratic programs. In particular, we modify the generation of the matrix $Q$: instead of using a diagonal structure, we replace it with a dense positive semidefinite matrix constructed as $Q = R^\top R$, where $R$ contains i.i.d. Gaussian entries. This removes the sparsity advantage typically leveraged by classical trust-region solvers and leads to substantially more challenging QP instances. The results are summarized in Table 7.

Table 7: Results on the convex QP task evaluated on the test set with 833 samples.

| Method | Max eq. | Max ineq. | sol. rel.err. | obj. rel.err. | Time (s) |
|---|---|---|---|---|---|
| Knitro | 0.0000 | 0.0000 | 0 | 0 | 0.0224 |
| osqp | 0.0000 | 0.0000 | $1.5 \times 10^{-3}$ | $1.8 \times 10^{-5}$ | 0.8020 |
| qpth | 0.0000 | 0.0000 | $1.5 \times 10^{-3}$ | $1.8 \times 10^{-5}$ | 0.0035 |
| DC3 | 0.0000 | 0.0000 | $5.2 \times 10^{-1}$ | $5.0 \times 10^{-1}$ | 0.0041 |
| **Descent** | 0.0000 | 0.0000 | $2.1 \times 10^{-2}$ | $9.3 \times 10^{-4}$ | 0.0131 |

Descent-Net continues to produce high-quality solutions under this more general setting, demonstrating that its effectiveness is not tied to diagonal or otherwise simplified structures.

## A.5 SCALABILITY EVALUATION ON PORTFOLIO OPTIMIZATION

A widely applicable instance of quadratic programming in real-world settings is the mean-variance portfolio optimization problem. The objective is to minimize portfolio risk while satisfying practical portfolio allocation constraints:

$$\min_{\mathbf{w}} \mathbf{w}^\top \Sigma \mathbf{w} \quad \text{s.t.} \quad \mathbf{w}^\top \mathbf{1} = 1, \ \mathbf{w}^\top \mu \geq R, \ \mathbf{w} \geq 0, \tag{16}$$

where $\mathbf{w}$ denotes asset weights, $\Sigma$ is the covariance matrix, $\mu$ is the expected return vector, and $R$ is the minimum return requirement.

We conduct portfolio optimization experiments with $n = 100, n = 800$, and $n = 4000$ assets to evaluate both the practical effectiveness and scalability of our method. For each problem size, we generate 10,000 synthetic benchmark instances. To emulate a market environment where asset co-movements evolve slowly, the covariance matrix is fixed across all instances and constructed as $\Sigma = A^\top A$, where entries of $A$ are sampled i.i.d. from a standard normal distribution. The expected return vectors $\mu$ are independently sampled from a uniform distribution over $[0, 1]$, modeling varying market conditions.

The return thresholds differ between training and testing. For training, each $R$ is drawn independently from a uniform distribution over $[0.05, 0.4]$. For testing, $R$ values are generated as a linearly spaced sequence over the same interval. We use a 9:1 train-test split.

The network contains a single hidden layer. Its width is set to 8 times the number of assets for the $n = 100$ experiment (i.e., 800), and 1.5 times the number of assets for the $n = 800$ and $n = 4000$ experiments (i.e., 1200 and 6000, respectively). The initial solution is refined using $S = 3$ descent updates for $n = 100$ and $S = 2$ descent updates for $n = 800$ and $n = 4000$. Both the Descent module and the step size $\beta$ are trained using Adam. The initial learning rates are $1 \times 10^{-3}$ for the Descent module, and 0.1, 0.1, and 0.01 for $\beta$ in the $n = 100$, $n = 800$, and $n = 4000$ experiments, respectively. Learning rates are decayed by a factor of 0.1 at epochs 100, 150, and 200 over a total of 300 epochs. All instances are initialized using the equal-weighted portfolio $w_i = 1/n$.

We compare Descent-Net with `osqp`, a widely used and highly optimized QP solver, as well as DC3 (Donti et al., 2021). All experiments were conducted on a server equipped with two AMD EPYC 9754 CPUs (128 cores each, 3.1 GHz) and an NVIDIA RTX 5090 GPU. The combined numerical results for all problem sizes are shown below.

Table 8: Test-set portfolio optimization results for $n = 100$ (batch size 512), $n = 800$ (batch size 100), and $n = 4000$ (batch size 10) assets

| $n = 100$ | ineq. vio. | eq. vio. | sol. rel. err. | obj. rel. err. | Time (s) |
|---|---|---|---|---|---|
| osqp | 0.0000 | 0.0000 | 0 | 0 | 0.0015 |
| DC3 | 0.0000 | 0.0000 | 2.8 | $5.4 \times 10^1$ | 0.0125 |
| Descent-Net | 0.0000 | 0.0000 | $1.4 \times 10^{-4}$ | $4.9 \times 10^{-6}$ | 0.0019 |

| $n = 800$ | ineq. vio. | eq. vio. | sol. rel. err. | obj. rel. err. | Time (s) |
|---|---|---|---|---|---|
| osqp | 0.0000 | 0.0000 | 0 | 0 | 0.0207 |
| Descent-Net | 0.0000 | 0.0000 | $9.3 \times 10^{-4}$ | $7.3 \times 10^{-6}$ | 0.0019 |

| $n = 4000$ | ineq. vio. | eq. vio. | sol. rel. err. | obj. rel. err. | Time (s) |
|---|---|---|---|---|---|
| osqp | 0.0000 | 0.0000 | 0 | 0 | 0.6024 |
| Descent-Net | 0.0000 | 0.0000 | $1.6 \times 10^{-4}$ | $1.2 \times 10^{-6}$ | 0.0044 |

We find that DC3 fails to produce feasible solutions for $n = 800$ and $n = 4000$ because its training diverges. This is likely due to DC3's reliance on gradient steps, which are used to enforce inequality constraints, but whose step sizes and momentum decay parameters are difficult to tune for large-scale settings. In contrast, Descent-Net remains accurate and highly efficient across all problem sizes.

The `osqp` times report the average runtime for a single instance, whereas the Descent-Net times correspond to the average runtime for a batch of instances. As shown, Descent-Net achieves lower runtimes while maintaining objective errors on the order of $10^{-6}$, demonstrating strong scalability to problems with thousands of variables.

## A.6 STEP SIZE SELECTION STRATEGIES

We compare the effectiveness of three different step size selection strategies:

- A fixed step size $\alpha = 1/M$;

- The maximum feasible step size $\alpha_{\max}$ that ensures feasibility;

- A learnable scale factor $\sigma(\beta)$ applied to $\alpha_{\max}$.

We perform comparative experiments on the convex quadratic program (13), evaluating three methods based on the feasibility and optimality of their solutions after a fixed number of update steps $S = 6$. The corresponding results are presented in Table 9. As shown, both the fixed step size $1/M$ and the maximum feasible step size $\alpha_{\max}$ perform worse than our final choice $\alpha = \sigma(\beta)\alpha_{\max}$. The limitation of $1/M$ lies in its lack of flexibility, as a fixed step size cannot adapt to the varying landscape of the problem. And $\sigma(\beta)\alpha_{\max}$ outperforms $\alpha_{\max}$ because the learnable parameter $\beta$ captures useful information that enables a more appropriate scaling of the maximum step size.

Table 9: Comparison of different step size selection strategies

| Method | ineq. vio. | eq. vio. | sol. rel. err. | obj. rel. err. |
|---|---|---|---|---|
| $1/M$ | 0.0000 | 0.0000 | $9.0 \times 10^{-2}$ | $1.2 \times 10^{-2}$ |
| $\alpha_{\max}$ | 0.0000 | 0.0000 | $1.1 \times 10^{-1}$ | $3.3 \times 10^{-2}$ |
| $\sigma(\beta)\alpha_{\max}$ | 0.0000 | 0.0000 | $1.4 \times 10^{-2}$ | $4.5 \times 10^{-4}$ |

### A.7 SUBPROBLEM

In our method, each descent direction $d_s$ is obtained by solving a subproblem (3). To assess the ability of the Descent-Net to solve this subproblem, we measure the relative error of the subproblem's objective value between each layer's output $d_k$ and the corresponding optimal solution.

We conduct experiments on the convex QP task. For simplicity, we set $S = 1$, performing only a single update, and fix the number of Descent-Net layers to $K = 3$. We then evaluate the trained network, with the results reported in Table 10. As shown, the objective value of the subproblem (Descent value) decreases progressively across layers, and by the final layer (layer 3), the relative error in the objective value has already been reduced to $0.001$, demonstrating the efficiency of the Descent-Net in solving the subproblem.

Table 10: Effectiveness of Descent-Net in solving subproblem

| Layer | Descent Value | Relative Error |
|---|---|---|
| 0 | 1740.4817 | 2.6463 |
| 1 | 505.2591 | 0.0585 |
| 2 | 478.2721 | 0.0020 |
| 3 | 477.7893 | 0.0010 |

### A.8 LEARNABLE $\gamma$ IN DESCENT-NET

We recorded the values of the learnable parameter $\gamma$ in each layer of the $S$ Descent Modules of the trained Descent-Net. For both QP and Nonconvex problems, $\gamma$ is initialized to $0.1$, while for the ACOPF problem it is initialized to $1$. The results are presented in Table 11 and Table 12. These results indicate that the network is able to adjust $\gamma$ dynamically across layers. In many cases, the values of $\gamma$ tend to decrease with the layer depth, which is consistent with the requirement of diminishing step sizes for convergence in subgradient methods.

### A.9 COMPARISON WITH PGM (PROJECTED SUBGRADIENT METHOD)

We compare Descent-Net with the original PGM. Specifically, we remove the operator $T^k$ in Descent-Net so that each layer reduces to (7). We still treat the step size $\gamma_k$ as a learnable parameter and train this degenerated network in the same manner as Descent-Net.

Table 11: Values of $\gamma$ in Descent-Net across steps for QP and Nonconvex problems

| | **QP** | | | | | |
|---|---|---|---|---|---|---|
| | Step 1 | Step 2 | Step 3 | Step 4 | Step 5 | Step 6 |
| $\gamma_1$ | $5.25 \times 10^{-3}$ | $9.25 \times 10^{-3}$ | $1.02 \times 10^{-2}$ | $9.20 \times 10^{-3}$ | $3.34 \times 10^{-2}$ | $6.59 \times 10^{-2}$ |
| $\gamma_2$ | $1.67 \times 10^{-2}$ | $2.90 \times 10^{-2}$ | $9.14 \times 10^{-3}$ | $8.12 \times 10^{-3}$ | $5.44 \times 10^{-3}$ | $2.64 \times 10^{-3}$ |
| $\gamma_3$ | $6.48 \times 10^{-2}$ | $3.61 \times 10^{-2}$ | $7.65 \times 10^{-3}$ | $3.12 \times 10^{-3}$ | $1.50 \times 10^{-3}$ | $1.09 \times 10^{-3}$ |
| | **Nonconvex** | | | | | |
| | Step 1 | Step 2 | Step 3 | Step 4 | Step 5 | Step 6 |
| $\gamma_1$ | $6.35 \times 10^{-3}$ | $9.47 \times 10^{-3}$ | $7.48 \times 10^{-3}$ | $1.35 \times 10^{-2}$ | $3.66 \times 10^{-2}$ | $9.13 \times 10^{-2}$ |
| $\gamma_2$ | $2.81 \times 10^{-2}$ | $2.47 \times 10^{-3}$ | $2.90 \times 10^{-3}$ | $4.60 \times 10^{-3}$ | $3.56 \times 10^{-3}$ | $3.44 \times 10^{-3}$ |
| $\gamma_3$ | $9.23 \times 10^{-2}$ | $3.09 \times 10^{-2}$ | $3.50 \times 10^{-3}$ | $2.96 \times 10^{-3}$ | $1.21 \times 10^{-3}$ | $1.18 \times 10^{-3}$ |

Table 12: Values of $\gamma$ in Descent-Net across steps for ACOPF problems

| | **node = 30, H-Proj** | | | | **node = 30, D-Proj** | | |
|---|---|---|---|---|---|---|---|
| | Step 1 | Step 2 | Step 3 | | Step 1 | Step 2 | Step 3 |
| $\gamma_1$ | 1.00 | 1.00 | 1.00 | $\gamma_1$ | 1.00 | 1.00 | 1.00 |
| $\gamma_2$ | 0.99 | 1.01 | 0.99 | $\gamma_2$ | 0.99 | 0.99 | 1.00 |
| $\gamma_3$ | 1.10 | 0.01 | 0.84 | $\gamma_3$ | 1.42 | 0.20 | 1.01 |
| | **node = 118, H-Proj** | | | | **node = 118, D-Proj** | | |
| | Step 1 | Step 2 | Step 3 | | Step 1 | Step 2 | Step 3 |
| $\gamma_1$ | 1.00 | 1.00 | 1.00 | $\gamma_1$ | 1.00 | 1.00 | 1.00 |
| $\gamma_2$ | 1.00 | 1.00 | 1.00 | $\gamma_2$ | 1.00 | 1.00 | 1.00 |
| $\gamma_3$ | 1.07 | 1.06 | 1.10 | $\gamma_3$ | 1.29 | 1.03 | 0.99 |

We evaluate the performance under different numbers of iterations $K$, with the results reported in Table 13. We observe that PGM is inefficient, as the relative error in the objective value compared to the initial solution decreases very little with increasing iterations. This is likely due to the difficulty of selecting an appropriate step size for PGM. In contrast, the Descent-Net achieves strong solution quality with only three layers, which also leads to a significant advantage in computational efficiency.

Table 13: Comparison of Descent-Net and PGM on the convex QP task.

| Method | ineq. vio. | eq. vio. | sol. rel. err. | obj. rel. err. | Time (s) |
|---|---|---|---|---|---|
| **PGM** ($K = 10$) | 0.0000 | 0.0000 | $1.9 \times 10^{-1}$ | $1.1 \times 10^{-1}$ | 0.0270 |
| **PGM** ($K = 20$) | 0.0000 | 0.0000 | $1.9 \times 10^{-1}$ | $1.1 \times 10^{-1}$ | 0.0501 |
| **PGM** ($K = 50$) | 0.0000 | 0.0000 | $1.9 \times 10^{-1}$ | $1.1 \times 10^{-1}$ | 0.1119 |
| **Descent** ($K = 3$) | 0.0000 | 0.0000 | $1.4 \times 10^{-2}$ | $4.5 \times 10^{-4}$ | 0.0152 |

### A.10  DESCENT UPDATES IN THE ACOPF PROBLEM

In the ACOPF problem, given $(n - m)$ entries of a feasible point $y \in \mathbb{R}^n$, the remaining $m$ entries are, in general, determined by the $m$ equality constraints $h_x(y) = 0$.

Following the method in Donti et al. (2021); Liang et al. (2024); Wu et al. (2025), we assume the existence of a function $\varphi_x : \mathbb{R}^{n-m} \to \mathbb{R}^m$ such that $h_x([z, \varphi_x(z)]) = 0$. This allows us to eliminate the equality constraints and reformulate the problem in terms of the partial variable $z$. We can then perform descent direction updates on $z$, where the optimization problem involves only the inequality constraints:

$$\min_{z \in \mathbb{R}^{n-m}} \tilde{f}_x(z), \quad \text{s.t.} \quad \tilde{g}_x(z) \leq 0, \tag{17}$$

where $\tilde{f}_x(z) = f_x\left([z^T, \varphi_x(z)^T]^T\right)$ and $\tilde{g}_x(z) = g_x\left([z^T, \varphi_x(z)^T]^T\right)$.

Using the chain rule, we can compute the derivative of $\varphi_x$ with respect to $z$, even without an explicit expression of $\varphi_x$:

$$0 = \frac{\mathrm{d}}{\mathrm{d}z} h_x\big(\varphi_x(z)\big) = \frac{\partial h_x}{\partial z} + \frac{\partial h_x}{\partial \varphi_x(z)} \frac{\partial \varphi_x(z)}{\partial z}$$

$$= J^h_{:,0:m} + J^h_{:,m:n} \frac{\partial \varphi_x(z)}{\partial z},$$

$$\Rightarrow \quad \frac{\partial \varphi_x(z)}{\partial z} = -\big(J^h_{:,m:n}\big)^{-1} J^h_{:,0:m}.$$

Here, $J^h \in \mathbb{R}^{m \times n}$ denotes the Jacobian matrix of the equality constraints $h_x(y)$ with respect to $y$. The notation $J^h_{:,0:m}$ and $J^h_{:,m:n}$ represents the submatrices corresponding to the partial derivatives with respect to $z$ and $\varphi_x(z)$, respectively.

From this result, we can further obtain the gradients of the objective and inequality constraints with respect to $z$. These gradient informations are then passed to the Descent-Net, which outputs the descent direction $d_z$ for the partial variable $z$.

In order to obtain the complete descent direction $d = [d_z, d_\varphi]$ for $y$, we also need the expression of $d_\varphi$. To ensure that the equality constraints remain satisfied, we require the following

$$h(z + \alpha d_z, \varphi(z) + \alpha d_\varphi) \approx h\big(z, \varphi(z)\big) + \alpha J^h \begin{bmatrix} d_z \\ d_\varphi \end{bmatrix}$$

$$= h\big(z, \varphi(z)\big) + \alpha\big(J^h_{:,0:m} d_z + J^h_{:,m:n} d_\varphi\big) = 0,$$

where $\alpha > 0$ is the step size. Hence, we obtain

$$d_\varphi = -\big(J^h_{:,m:n}\big)^{-1} J^h_{:,0:m} d_z \; - \; \big(J^h_{:,m:n}\big)^{-1} \frac{h\big(z, \varphi(z)\big)}{\alpha}.$$

## A.11 PROOF OF PROPOSITION 1

*Proof.* Given any vector $d \in \mathbb{R}^n$, we aim to compute its projection onto $\mathcal{D}$, i.e., solve the following problem:

$$\min_{d' \in \mathbb{R}^n} \frac{1}{2} \|d' - d\|_2^2 \quad \text{s.t.} \quad \|d'\|_2 \leq 1, \quad H^\top d' = 0.$$

Without loss of generality, we assume that the Linear Independence Constraint Qualification (LICQ) holds. Otherwise, the projection reduces to the origin $d' = 0$. Now we derive the KKT conditions for this problem from the Lagrangian

$$\mathcal{L}(d', \lambda, \mu) = \frac{1}{2}\|d' - d\|_2^2 + \lambda^\top H^\top d' + \mu(\|d'\|_2^2 - 1),$$

where $\lambda \in \mathbb{R}^{n-m}$ and $\mu \geq 0$ are the Lagrange multipliers.

Taking the gradient with respect to $d'$ and setting it to zero gives:

$$d' - d + H\lambda + 2\mu d' = 0 \quad \Rightarrow \quad (1 + 2\mu)d' + H\lambda = d.$$

Since $H^\top d' = 0$, we have

$$H^\top H \lambda = (1 + 2\mu)H^\top d' + H^\top H \lambda = H^\top d \quad \Rightarrow \quad \lambda = (H^\top H)^{-1} H^\top.$$

We consider two cases:

**Case 1:** If $\mu = 0$, then the projection is

$$d' = d - H\lambda = d - H(H^\top H)^{-1} H^\top d = \hat{d}.$$

**Case 2:** If $\mu > 0$, then we have $\|d'\| = 1$ and

$$(1 + 2\mu)^2 = (1 + 2\mu)^2 (d')^\top d' = (d - H\lambda)^\top (d - H\lambda) = \hat{d}^\top \hat{d} = \|\hat{d}\|^2.$$

Hence, the projection is:

$$d' = \frac{1}{1 + 2\mu} \left( d - H(H^\top H)^{-1} H^\top d \right) = \frac{1}{\|\hat{d}\|} \hat{d}.$$

$\square$

## A.12 PROOF OF THEOREM 4.1

The following result is standard for projected subgradient method for solving convex problems.

**Lemma 2.** *For any $\varepsilon > 0$, there exists a constant $C > 0$ such that if we set $K = \frac{C}{\varepsilon^2}$ and choose the step size in (7) as $\gamma_k = \frac{1}{\sqrt{K}}$, then*

$$\min_{1 \leq k \leq K} g(d_k) - g(d^*) \leq \varepsilon,$$

*where $d^*$ denotes the optimal solution of Problem (3).*

*Proof.* Let $\mathbf{u}_k = \nabla f_x(y) + \sum_{j=1}^{l} c_j \mathbf{1}_{\{\langle d^k, \nabla g_{x,j}(y) \rangle \geq -M g_{x,j}(y)\}} \nabla g_{x,j}(y)$, and define $G = \|\nabla f_x(y)\| + \sum_{j=1}^{l} c_j \|\nabla g_{x,j}(y)\| < \infty$. Then it follows that $\|\mathbf{u}_k\|^2 \leq G^2$ for all $k$.

By the non-expansiveness (contractive property) of the projection operator, we have:

$$\begin{aligned}
\|d_{k+1} - d^*\|^2 &= \|\mathcal{P}(\tilde{d}_{k+1}) - \mathcal{P}(d^*)\|^2 \\
&\leq \|\tilde{d}_{k+1} - d^*\|^2 \\
&= \|d_k - \gamma_k \mathbf{u}_k - d^*\|^2 \\
&= \|d_k - d^*\|^2 - 2\gamma_k \langle \mathbf{u}_k, d_k - d^* \rangle + \gamma_k^2 \|\mathbf{u}_k\|^2.
\end{aligned}$$

Moreover, since $\mathbf{u}_k \in \partial g(d_k)$ and $g$ is convex, we have

$$g(d_k) - g(d^*) \leq \langle \mathbf{u}_k, d_k - d^* \rangle.$$

Substituting this into the previous inequality gives:

$$\|d_{k+1} - d^*\|^2 \leq \|d_k - d^*\|^2 - 2\gamma_k(g(d_k) - g(d^*)) + \gamma_k^2 \|\mathbf{u}_k\|^2$$
$$\leq \|d_k - d^*\|^2 - 2\gamma_k(g(d_k) - g(d^*)) + \gamma_k^2 G^2.$$

Rearranging and summing both sides from $k = 1$ to $K$, we get

$$\sum_{k=1}^{K} \gamma_k(g(d_k) - g(d^*)) \leq \frac{1}{2} \left( \|d_1 - d^*\|^2 - \|d_{K+1} - d^*\|^2 + G^2 \sum_{k=1}^{K} \gamma_k^2 \right)$$
$$\leq \frac{1}{2}(\|d_1 - d^*\|^2 + G^2),$$

we let $C = \sqrt{(\|d_1 - d^*\|^2 + G^2)/2}$.

On the other hand, we have

$$\sum_{k=1}^{K} \gamma_k(g(d_k) - g(d^*)) = \frac{1}{\sqrt{K}} \sum_{k=1}^{K} (g(d_k) - g(d^*))$$
$$\geq \frac{1}{\sqrt{K}} \sum_{k=1}^{K} \left( \min_{1 \leq k \leq K} g(d_k) - g(d^*) \right)$$
$$= \sqrt{K} \left( \min_{1 \leq k \leq K} g(d_k) - g(d^*) \right).$$

Combining both inequalities, we obtain:

$$\min_{1 \leq k \leq K} g(d_k) - g(d^*) \leq \frac{1}{\sqrt{K}} \cdot \frac{1}{2} \left( \|d_1 - d^*\|^2 + G^2 \right) = \frac{\varepsilon}{\sqrt{C}} \cdot \sqrt{C} = \varepsilon.$$

$\square$

**Lemma 3.** *Given the sequence of iterates $\{d_1^{proj}, \cdots, d_K^{proj}\}$ generated by the projected gradient method (7) with initial input $d_0$, there exists a $K$-layer Descent-Net with a specific parameter assignment that, starting from the same initial input $d_0$, it produces the same iterative sequence, i.e., $d_k = d_k^{proj}$ for all $1 \leq k \leq K$.*

*Proof.* It suffices to show that there exists a set of parameters such that $T^k(\mathbf{u}_k) = \gamma_k \mathbf{u}_k$ for all $1 \leq k \leq K$, where $\mathbf{u}_k$ is defined in (8).

Let $\mathbf{W}^k \in \mathbb{R}^{q \times n}$ be a full column rank matrix , so its left pseudo-inverse $(\mathbf{W}^k)^\dagger \in \mathbb{R}^{n \times q}$ exists and satisfies $(\mathbf{W}^k)^\dagger \mathbf{W}^k = I_n$.

By assuption (4) and the defination of $\mathbf{u_k}$, we have

$$\|\mathbf{u}_k\|_2 \leq \|\nabla f_x(y)\|_2 + \sum_{j=1}^{l} |c_j| \cdot \sqrt{n} \cdot |\nabla g_{x,j}(y)|$$
$$\leq \|\nabla f_x(y)\|_2 + \max_j(c_j) \cdot \sqrt{n} \cdot \|\nabla g_x(y)\|_1$$
$$\leq \|\nabla f_x(y)\|_2 + \max_j(c_j) \cdot n \cdot \|\nabla g_x(y)\|_2$$
$$\leq L_f + \max_j(c_j) \cdot n \cdot L_g$$

In the derivation of the third inequality, we used the equivalent norm theorem. Then we have

$$\|\mathbf{W}^k \mathbf{u}_k\|_1 \leq \sqrt{n}\|\mathbf{W}^k \mathbf{u}_k\|_2 \leq \sqrt{n}\|\mathbf{W}^k\|_2 \|\mathbf{u}_k\|_2 \leq \sqrt{n}\|\mathbf{W}^k\|_2 (L_f + \max_j(c_j) \cdot n \cdot L_g)$$

Let $L = \sqrt{n}\|\mathbf{W}^k\|_2(L_f + \max_j(c_j) \cdot n \cdot L_g)$. Define the bias vector as $\mathbf{b}_1^k = L \cdot \mathbf{1}_q$, where $\mathbf{1}_q$ denotes the $q$-dimensional vector with all entries equal to one. Then we have

$$\mathrm{ReLU}(\mathbf{W}^k\mathbf{u}_k + \mathbf{b}_1^k) = \mathbf{W}^k\mathbf{u}_k + \mathbf{b}_1^k,$$

since each coordinate of $\mathbf{W}^k\mathbf{u}_k + \mathbf{b}_1^k$ is positive.

Now, let the second layer weight be $\mathbf{V}^k = \gamma_k(\mathbf{W}^k)^\dagger$, and the second bias be $\mathbf{b}_2^k = -\gamma_k(\mathbf{W}^k)^\dagger\mathbf{b}_1^k$. Then we have:

$$T^k(\mathbf{u}_k) = \mathbf{V}^k \cdot \mathrm{ReLU}(\mathbf{W}^k\mathbf{u}_k + \mathbf{b}_1^k) + \mathbf{b}_2^k = \gamma_k(\mathbf{W}^k)^\dagger(\mathbf{W}^k\mathbf{u}_k + \mathbf{b}_1^k) - \gamma_k(\mathbf{W}^k)^\dagger\mathbf{b}_1^k = \gamma_k\mathbf{u}_k.$$

This completes the proof. $\qquad\square$

We now proceed to prove Theorem 4.1. First, by Lemma 2, we know that for any $\varepsilon > 0$, by choosing an appropriate step size, there exists an iteration sequence $\{d_k^{\mathrm{proj}}\}_{k=1}^K$ generated by the projected subgradient method such that

$$\left|\min_{1 \le k \le K} g(d_k^{\mathrm{proj}}) - g(d^*)\right| < \varepsilon.$$

Let $K_\varepsilon = \mathrm{argmin}_{1 \le k \le K}\, g(d_k^{\mathrm{proj}})$, then we have $K_\varepsilon \le K = C/\varepsilon^2$ and

$$|g(d_{K_\varepsilon}^{\mathrm{proj}}) - g(d^*)| < \varepsilon.$$

Moreover, by Lemma 3, we know that there exists there exists a $K_\varepsilon$ layer Descent-Net such that the output of each layer exactly matches the corresponding iterate sequence $\{d_k^{\mathrm{proj}}\}_{k=1}^{K_\varepsilon}$. In particular, we have

$$d_{K_\varepsilon} = d_{K_\varepsilon}^{\mathrm{proj}}$$

Therefore,

$$\|g(d_{K_\varepsilon}) - g(d^*)\| < \varepsilon,$$

which is exactly the desired result.

### A.13 PROOF OF LEMMA 1

To prove Lemma 1, for convenience, we have the following new notations.

- Gradient of the linearized objective: $p := \nabla f_x(y) \in \mathbb{R}^n$.
- Active–constraint data (for $j = 1, \ldots, l$): $a_j := \nabla g_{x,j}(y) \in \mathbb{R}^n$ and $b_j := -M\, g_j(y)$.
- Linear-equality matrix: $E := \nabla h_x(y)^\top \in \mathbb{R}^{m \times n}$. Write $P := I - E^\top(EE^\top)^{-1}E$ for the orthogonal projector onto $\ker(E)$.
- Search set (unit Euclidean ball in the null–space of $E$):
$$\mathcal{D} := \{\, d \in \mathbb{R}^n \mid Ed = 0,\ \|d\|_2 \le 1 \,\}.$$

- Feasible set of Topkis–Veinott UFD sub-problem with $l_2$ norm constraint:
$$F := \{\, d \in \mathcal{D} \mid \langle a_j, d\rangle \le b_j,\ j = 1, \ldots, l \,\}.$$

Using these notations, problem (3) can be written as

$$\min_{d \in \mathcal{D}}\ \Phi(d) := \langle p, d\rangle + \sum_{j=1}^l c_j \max\{\langle a_j, d\rangle,\, b_j\}, \qquad c_j > 0. \tag{Pen}$$

We will it is equivalent to the following constrained problem with appropriate $c_j$:

$$\min_{d \in F}\ \langle p, d\rangle. \tag{UFD-L2}$$

We now prove Lemma 1 in the main context. With our new notation, we rewrite it as the following lemma.

**Lemma 4** (Exact hinge penalty on an $\ell_2$–ball). *Let $c_{\min} = \min_j c_j$. If we have*

$$c_{\min} > \frac{L_f}{M\delta_g}, \tag{18}$$

*then every global minimizer of (Pen) is feasible for problem (UFD-L2), hence*

$$\arg\min_{d \in F} \langle p, d \rangle \;=\; \arg\min_{d \in \mathcal{D}} \Phi(d).$$

*Proof.* Let

$$\tilde{L} := \|\nabla f_x(y)\|_2, \qquad b_{\min} := \min_{j : b_j > 0} b_j.$$

By assumption (4) and (5), we have $\tilde{L} \le L_f$, $b_{\min} \ge M\delta_g$. Hence

$$c_{\min} > \frac{L_f}{M\delta_g} \ge \frac{\tilde{L}}{b_{\min}}$$

Suppose $d \in \mathcal{D}$ is the optimal point of problem (Pen), but $d \notin F$. Define the violation vector $r(d) := \left( \left[ \langle a_1, d \rangle - b_1 \right]_+, \dots, \left[ \langle a_l, d \rangle - b_l \right]_+ \right) \in \mathbb{R}^l_{\ge 0}$. Let $V(d) := \{ j \mid r_j(d) > 0 \}$ be the index set of violated constraints.

If $r(d) = 0$ then $d \in F$. Otherwise put

$$\alpha(d) := \min_{j \in V(d)} \frac{b_j}{\langle a_j, d \rangle} \in (0, 1), \qquad \hat{d} := \alpha(d)\, d.$$

Since $Ed = 0$ and $\alpha(d) \le 1$, one has $\hat{d} \in \mathcal{D}$. Moreover, for every $j$, $\langle a_j, \hat{d} \rangle = \alpha(d)\langle a_j, d \rangle \le b_j$, so $\hat{d} \in F$.

Pick $\bar{j} \in V(d)$ that attains the minimum in $\alpha(d)$ and set $\delta := r_{\bar{j}}(d) = \langle a_{\bar{j}}, d \rangle - b_{\bar{j}} > 0$. Then

$$1 - \alpha(d) = 1 - \frac{b_{\bar{j}}}{\langle a_{\bar{j}}, d \rangle} = \frac{\delta}{\langle a_{\bar{j}}, d \rangle} \le \frac{\delta}{b_{\bar{j}}} \le \frac{\delta}{b_{\min}},$$

where we use $\langle a_{\bar{j}}, d \rangle > b_{\bar{j}}$. Because $\|d\|_2 \le 1$, we obtain

$$\|d - \hat{d}\|_2 = (1 - \alpha(d))\|d\|_2 \le \frac{\delta}{b_{\min}}.$$

First, the linear part is $\tilde{L}$-Lipschitz on $\mathcal{D}$:

$$|\langle p, d \rangle - \langle p, \hat{d} \rangle| \le \tilde{L}\|d - \hat{d}\|_2 \le \frac{\tilde{L}}{b_{\min}}\,\delta.$$

Next, because $\hat{d} \in F$ we have $\max\{\langle a_j, \hat{d} \rangle, b_j\} = b_j$ for every $j$, whereas for $d$

$$\max\{\langle a_j, d \rangle, b_j\} - b_j = \left[ \langle a_j, d \rangle - b_j \right]_+ = r_j(d).$$

Hence

$$\Phi(d) - \Phi(\hat{d}) = \left( \langle p, d \rangle - \langle p, \hat{d} \rangle \right) + \sum_{j \in V(d)} c_j r_j(d).$$

The second term is bounded below by $c_{\min} \sum_{j \in V(d)} r_j(d) \ge c_{\min}\delta$, so using the Lipschitz bound,

$$\Phi(d) - \Phi(\hat{d}) \ge \left( c_{\min} - \frac{\tilde{L}}{b_{\min}} \right)\delta.$$

By definition, the coefficient of $\delta$ is positive, hence $\Phi(d) > \Phi(\hat{d})$ for $\hat{d} \in \mathcal{D}$, which contradicts with the optimality. Therefore all global minimizers of (Pen) lie in $F$.

On $F$ the penalty term vanishes, i.e. $\Phi(d) = \langle p, d \rangle + \sum_j c_j b_j$. Thus (UFD-L2) and (Pen) share the same minimizers and their optimal values differ only by the constant $\sum_j c_j b_j$. $\qquad\square$

### A.14 PROOF OF THEOREM 4.2

**Definition 1** (Fritz–John point). *Let $y \in \mathbb{R}^n$ be a feasible point for the problem*

$$\min f(y) \quad \text{s.t.} \quad h_i(y) = 0, \ g_j(y) \leq 0.$$

*Then $y$ is called a* Fritz–John point *if there exist multipliers $\lambda_0 \geq 0$, $\lambda_j \geq 0$ for $j = 1, \ldots, l$, and $\mu_i \in \mathbb{R}$ for $i = 1, \ldots, m$, not all zero, such that*

$$\lambda_0 \nabla f(y) + \sum_{j=1}^{l} \lambda_j \nabla g_j(y) + \sum_{i=1}^{m} \mu_i \nabla h_i(y) = 0,$$

$$\lambda_j \cdot g_j(y) = 0, \quad j = 1, \ldots, l.$$

**Definition 2** (KKT point). *A feasible point $y \in \mathbb{R}^n$ is called a* Karush–Kuhn–Tucker (KKT) point *if there exist multipliers $\lambda_j \geq 0$ and $\mu_i \in \mathbb{R}$ such that*

$$\nabla f(y) + \sum_{j=1}^{l} \lambda_j \nabla g_j(y) + \sum_{i=1}^{m} \mu_i \nabla h_i(y) = 0,$$

$$\lambda_j \cdot g_j(y) = 0, \quad j = 1, \ldots, l.$$

If the LICQ condition holds at $\bar{y}$, then the Fritz-John point is also a KKT point.

**Lemma 5** (Farkas Lemma with equality constraints). *Let $A \in \mathbb{R}^{m \times n}$, $B \in \mathbb{R}^{p \times n}$, and $b \in \mathbb{R}^m$.*

*Then exactly one of the following two systems has a solution:*

    *(a) There exists $x \in \mathbb{R}^n$ such that*
$$Ax < b, \quad Bx = 0.$$

    *(b) There exists $(\lambda, \mu) \in \mathbb{R}^m \times \mathbb{R}^p$, not both zero, such that*
$$\lambda \geq 0, \quad A^\top \lambda + B^\top \mu = 0, \quad \lambda^\top b \leq 0.$$

*Moreover, both systems cannot be simultaneously feasible.*

We show that the problem (UFD-L2) has negative value if $y$ is not a Fritz-John point.

**Lemma 6** (Descent direction under failure of Fritz–John). *Let $\bar{y} \in \mathcal{C}$ be a feasible point. Suppose that the set of vectors*

$$\left\{ v = \lambda_0 \nabla f_x(\bar{y}) + \sum_{j=1}^{l} \lambda_j \nabla g_j(\bar{y}) + \sum_{i=1}^{m} \mu_i \nabla h_i(\bar{y}) \ \middle| \ \lambda_0 \geq 0, \ \lambda_j \geq 0, \ (\lambda, \mu) \neq 0, \ \lambda_j g_j(\bar{y}) = 0 \right\}$$

*does not contain the zero vector. That is, $\bar{y}$ is not a Fritz–John point.*

*Then there exists a vector $d \in \mathbb{R}^n$ such that:*

    • $\nabla h_x(\bar{y})^\top d = 0$,

    • $\nabla f_x(\bar{y})^\top d < 0$,

    • $\nabla g_j(\bar{y})^\top d < -M g_j(\bar{y})$ for all $j = 1, \ldots, l$,

    • $\|d\|_2 \leq 1$.

*Proof.* Let $H := [\nabla h_1(\bar{y}), \ldots, \nabla h_m(\bar{y})] \in \mathbb{R}^{n \times m}$, and define the subspace of directions satisfying the linearised equality constraints:

$$\mathcal{T} := \left\{ d \in \mathbb{R}^n \mid H^\top d = 0 \right\}.$$

Let $A \in \mathbb{R}^{(l+1) \times n}$ be the matrix whose rows are:

$$
A := \begin{bmatrix} \nabla f_x(\bar{y})^\top \\ \nabla g_1(\bar{y})^\top \\ \vdots \\ \nabla g_l(\bar{y})^\top \end{bmatrix}, \qquad b := \begin{bmatrix} 0 \\ -M g_1(\bar{y}) \\ \vdots \\ -M g_l(\bar{y}) \end{bmatrix}.
$$

Then we consider the system:

$$
Ad < b, \quad \text{subject to } H^\top d = 0.
$$

Since $\bar{y}$ is not a Fritz–John point, the system of equalities

$$
\lambda_0 \nabla f_x(\bar{y}) + \sum_j \lambda_j \nabla g_j(\bar{y}) + \sum_i \mu_i \nabla h_i(\bar{y}) = 0 \quad \text{with } \lambda_0 \geq 0,\ \lambda_j \geq 0, \mu_i \text{ not all zero,}
$$

has no solution satisfying the complementarity condition $\lambda_j g_j(\bar{y}) = 0$.

Therefore, by the Farkas Lemma 5, the dual system:

$$
\text{find } d \in \mathcal{T} \text{ such that } Ad < b
$$

is feasible.

Because $A, b$ are fixed and $b_j = -M g_j(\bar{y}) \geq 0$, which is finite for all $j = 1, \ldots, l$, and $\mathcal{T}$ is a linear subspace, the feasible set is convex and open in $\mathcal{T}$. We can scale $d$ such that $\|d\|_2 \leq 1$.

Hence, such a direction $d$ exists satisfying the claimed conditions. $\qquad \square$

By Lemma (1) and Theorem 4.1, if the following algorithm—based on the subproblem (Pen)—converges subsequently to a KKT point, then there exist constants $S > 0$ and $K > 0$, and certain network parameters $\Theta := \{\mathbf{V}^k, \mathbf{W}^k, \mathbf{b}_1^k, \mathbf{b}_2^k\}_{k=0,1,\ldots,K-1}$ and $\{\beta_s\}_{s=0,\ldots,S-1}$, such that **Descent-Net** returns a KKT point of the original problem (1). We therefore begin by proving the convergence of the algorithm stated below.

**Algorithm (UFD–penalty method).** Given a feasible starting point $y_0 \in \mathcal{C}$, repeat for $k = 0, 1, \ldots$

1. With the condition (4) holds, solve the sub–problem (Pen) at the current iterate $y_k$ and obtain a minimizer $d_k$.

2. Update $\quad y_{k+1} = y_k + \alpha_k d_k$, where $\alpha_k := \arg\min_\alpha \{f_x(y_k + \alpha d_k) \mid \alpha \in (0, 1/M]\}$, (*Lemma 1 implies* $y_{k+1} \in \mathcal{C}$)

We have the following result, which is similar to the Topkis–Veinott method Zoutendijk (1960); Faigle et al. (2013).

**Theorem A.1** (global convergence of the UFD–$L_2$ method). *Suppose the Assumption 1 2 and 3 hold. Furthermore, we assume that the gradient $\nabla f_x(y)$ is $L$−Lipschitz continuous, and $h_x, g_x$ are linear. Then every accumulation point $\bar{y}$ of the sequence $\{y_k\}$ generated by the UFD-penalty algorithm satisfies the KKT conditions of the problem (1).*

*Proof.* **(i) Feasibility and boundedness.** Lemma 4 shows that every $d_k$ satisfies $\nabla g_j(y_k)^\top d_k \leq -M g_j(y_k)$, hence $g_j(y_{k+1}) = g_j(y_k) + \alpha \nabla g_j(y_k)^\top d_k \leq 0$, where $\alpha \in (0, 1/M]$. Equality constraints are preserved by $\nabla h_x(y_k)^\top d_k = 0$, so $y_{k+1} \in \mathcal{C}$. Because $\{y \in \mathcal{C} \mid f_x(y) \leq f_x(y_0)\}$ is bounded, $\{y_k\}$ is bounded and admits convergent subsequences.

**Step (ii): every accumulation point is a Fritz–John point.**

Let $\bar{y}$ be an accumulation point of $\{y_k\}$, extracted from a subsequence $\{y_{k_s}\}$. Suppose, for contradiction, that $\bar{y}$ is not a Fritz–John point.

Then, by Lemma 6, there exist $z < 0$ and a direction $d \in \mathbb{R}^n$ satisfying:

$$\|d\|_2 \leq 1, \quad \nabla h_x(\bar{y})^\top d = 0, \quad \nabla f_x(\bar{y})^\top d < z < 0, \quad \nabla g_j(\bar{y})^\top d < -Mg_j(\bar{y}) + z.$$

Since $f_x$, $g_j$, and all gradients are continuous, and $y_{k_s} \to \bar{y}$, there exists $\varepsilon > 0$ and $\delta > 0$ such that for all $s$ sufficiently large (i.e., $\|y_{k_s} - \bar{y}\| < \delta$):

$$\nabla f_x(y_{k_s})^\top d < z + \varepsilon,$$
$$\nabla g_j(y_{k_s})^\top d < -Mg_j(y_{k_s}) + \varepsilon,$$
$$\nabla h_x(y_{k_s})^\top d < \varepsilon.$$

Fix $\varepsilon := |z|/3 > 0$. Then for large $s$, we obtain:

$$\nabla f_x(y_{k_s})^\top d < z + \varepsilon =: \hat{z} < 0,$$
$$\nabla g_j(y_{k_s})^\top d < -Mg_j(y_{k_s}) + \varepsilon,$$
$$\nabla h_x(y_{k_s})^\top d < \varepsilon.$$

Now consider the solution $d_{k_s}$ of the UFD subproblem (UFD-L2) at $y_{k_s}$, which satisfies:

$$\|d_{k_s}\|_2 \leq 1, \quad \nabla h_x(y_{k_s})^\top d_{k_s} = 0, \quad \nabla g_j(y_{k_s})^\top d_{k_s} \leq -Mg_j(y_{k_s}).$$

Since $d$ is a feasible direction and $\nabla f_x(y_{k_s})^\top d < \hat{z} < 0$, it follows that the optimal value $z_s := \nabla f_x(y_{k_s})^\top d_{k_s}$ must also satisfy:

$$z_s < \hat{z} < 0.$$

Thus, for all large $s$, we have:

$$\nabla f_x(y_{k_s})^\top d_{k_s} = z_s < 0, \quad \nabla g_j(y_{k_s})^\top d_{k_s} < 0, \quad \nabla h_x(y_{k_s})^\top d_{k_s} = 0.$$

Now define $y_{k_{s+1}} := y_{k_s} + td_{k_s}$, where $t > 0$ is small. Since $d_{k_s}$ satisfies the linearized equality constraints exactly and inequality constraints strictly, Taylor expansion gives:

$$f_x(y_{k_s} + td_{k_s}) = f_x(y_{k_s}) + t\nabla f_x(y_{k_s})^\top d_{k_s} + o(t) < f_x(y_{k_s}) + tz_s/2,$$
$$g_j(y_{k_s} + td_{k_s}) = g_j(y_{k_s}) + t\nabla g_j(y_{k_s})^\top d_{k_s} + o(t) < 0,$$
$$h_i(y_{k_s} + td_{k_s}) = h_i(y_{k_s}) + t\nabla h_i(y_{k_s})^\top d_{k_s} = 0.$$

Therefore, for sufficiently small $t > 0$, the updated point $y_{k_{s+1}} := y_{k_s} + td_{k_s}$ remains feasible and decreases the objective value.

**Contradiction conclusion.**

This contradicts the assumption that $\{f_x(y_k)\}$ converges to a finite value (since it would go to $-\infty$). Hence, our assumption must be false: every limit point $\bar{y}$ must satisfy the Fritz–John condition.

**(iii) LICQ $\Rightarrow$ KKT.** Under LICQ the Fritz–John multipliers have $\lambda_0 > 0$, so the KKT system holds at $\bar{y}$. $\qquad\square$

