# OpenReview forum: "Descent-Net: Learning Descent Directions for Constrained Optimization"
_ICLR.cc/2026/Conference — ICLR 2026 Conference Withdrawn Submission_

### Official Review · Reviewer_6Lpd · 2025-10-31

**Soundness:** 1
**Presentation:** 2
**Contribution:** 1
**Rating:** 0
**Confidence:** 5

**Summary:**

This paper proposes a machine learning architecture for constrained optimization learning that approximates an iterative descent algorithm. The proposed approach integrates an active set strategy, an approximate descent direction computation, and a projection operator to ensure equality constraint feasibility. The approach is evaluated on small nonlinear problems (synthetic convex QPs, a mildly nonlinear variant of these, and small AC Optimal Power Flow instances).

Overall, I have several reservations regarding the validity of the proposed methodology, stemming from limited assumptions (e.g. linear equality constraints) and unpractical existential results (universal approximation theorem). In particular, as far as I can tell, the proposed scheme, in general, is not guaranteed to converge to a feasible solution. Furthermore, numerical experiments are conducted on small instances that are either synthetic or orders of magnitude smaller than real-life instances.

**Strengths:**

* The paper considers a constrained learning task (where the output of a machine learning model should satisfy constraints), which has received less attention in the literature (although there is a growing body of works in that field)
* The paper leverages insights from existing optimization algorithms to inform the design of their ML framework

**Weaknesses:**

* The proposed method relies on inverting the matrix $H^{\top}H$, where $H$ is the Jacobian of equality constraints at the current iterate. This is a large obstacle to scalability, as i) this matrix may become dense and numerically ill-conditioned (especially close to the optimum), and (ii) forming and inverting this matrix will become expensive for larger instances.

* The convergence result of Theorem 4.2 relies on the universal approximation theorem of ReLU networks. It is an existential result, which only states that there exists an architecture and parameter assignment such that the returned point is a KKT point. It does not provide any practical guarantee because one can never know whether the network architecture is large enough.

* In addition, the universal approximation theorem is only valid for functions with bounded Lipschitz constant, which is not the case for the case at hand: the problem setup only assumes that objective and constraint functions $f_{x}(y), g_{x}(y), h_{x}(y)$ are smooth functions of $y$ given $x$. Additional assumptions would be needed regarding the domain of $x$ and the smoothness of $f, g, h$ w.r.t $x$.

    (note that even with those additional assumptions, the result of Theorem 4.2 remains impractical)

* The step size rule makes the following restrictive assumption (Section 4.3): "We assume that all equality constraints are linear."

* The step size rule relies on a local linear approximation of the inequality constraints using a first-order Taylor expansion. The proposed rule for controlling the step size also relies on that linear approximation, and may therefore fail to ensure feasibility in general.

* Numerical experiments are reported on small synthetic instances (which are not representative of real-life problems) and small AC-OPF instances. Regarding the latter, results should be reported on AC-OPF instances with several thousands of buses. There are several publicly-available datasets for this, e.g., [PGLearn](https://huggingface.co/PGLearn), which has data for systems with up to 24,000 buses

**Questions:**

* Problem MFD is formulated with an $\ell_{\infty}$ constraint on $d$, problem UFD is formulated with an $\ell_{1}$ box constraint on $d$, and problem (3) is formulated with an $\ell_{2}$ box constraint on $d$. Can you please explain why the different treatment in Eq. (3), and the potential impact of using an $\ell_{2}$ box instead of $\ell_{1}$ of $\ell_{\infty}$?
* Please comment on the validity of the linear approximation in the step size computation of Sec. 4.3, especially regarding feasibility guarantees
* I am very surprised by the computing times reported for ACOPF instances. According to [this benchmark](https://discourse.julialang.org/t/ac-optimal-power-flow-in-various-nonlinear-optimization-frameworks/78486/98), solving the AC-OPF takes about 0.1s and 0.3s for the 30 and 118 bus systems, respectively.
  I understand that Table 3 reports average solution times, but taking into account that i) these cases can be solved 4x faster than reported and ii) one GPU costs about 100x more than a CPU, the resulting speedup is actually quite modest compared to CPU-only solvers

---

> ### Author Response · Authors · 2025-11-23
>
> We thank the reviewer for their constructive feedback and for highlighting the contributions of our work:
> 1. The paper tackles the challenging and relatively less-explored problem of learning under hard constraints.
> 2. The method is principled, drawing on insights from classical optimization.
>
> We also summarize the reviewer’s main concerns:
> 1. The projection in our method may be computationally expensive — however, in many practical cases our method has lower complexity.
> 2. The reviewer suggests our convergence result depends on the universal approximation theorem, which is not the case.
> 3. The step-size rule may rely too strongly on linearity assumptions.
> 4. The reviewer questions the practical improvement of our method on ACOPF benchmarks. To address this we have improved the procedure for obtaining initial points, which further enhances our method's performance.
>
> We now address each point in detail below.
>
>
> ## Weakness 1
>
> We thank the reviewer for pointing out this concern. We clarify a few points below regarding the inversion of $H^\top H$, and we have also added the corresponding explanation in Proposition 1 of the paper.
>
> 1. In many practical applications, such as decision-focused learning(Tan et al., 2020), the equality-constraint Jacobian $H$ remains constant across problem instances. In such cases, $(H^\top H)^{-1}$ can be computed once during initialization and reused for all subsequent iterations.
>
> 2. Even for nonlinear problems where $H$ varies, each iteration of our method only requires $O(m^3)$ operations to compute the projection, and the total number of iterations is $S \cdot K$ (number of descent steps times network layers). In contrast, classical interior-point or trust-region solvers must solve a Newton system of size $n+m$ at each iteration, which costs $O((n+m)^3)$. Since in most applications $n > m$, this leads to at least a **8×** decrease in computational complexity.
>
> **References**
> - Yingcong Tan, Daria Terekhov, and Andrew Delong. *Learning linear programs from optimal decisions*. Advances in Neural Information Processing Systems, 33:19738–19749, 2020.
>
>
> ## Weakness 2 & 3
>
> We appreciate the reviewer’s insightful comment. We clarify that the convergence result in Theorem 4.2 **does not rely on** the universal approximation theorem. Instead, the proof of Theorem 4.2 uses the result of Lemma 3 in Appendix A.10.
>
> In Lemma 3, the existence of a network that satisfies the required properties is established through a **constructive proof**, rather than invoking the universal approximation theorem. Therefore, the concern regarding the network’s existence or Lipschitz assumptions is not relevant.
>
> However, we fully agree that the current guarantee remains existential and does not tell us how large the network should be to ensure convergence in practice. This is a challenging issue, and addressing it will be an important direction for future work.
>
> ## Weakness 4
>
> We note that linear constraints are common in many practical optimization problems, particularly in real-world decision-making tasks. A representative example is the widely studied **mean-variance portfolio optimization** problem in finance.
>
> Specifically, the optimization objective is to minimize portfolio risk measured by covariance, while satisfying practical allocation constraints:
>
> $$
> \min_{\mathbf{w}} \ \mathbf{w}^\top \Sigma \mathbf{w}
> \quad \text{s.t.} \quad
> \mathbf{w}^\top \textbf{1} = 1,\;
> \mathbf{w}^\top \mu \ge R,\;
> \mathbf{w} \ge 0,
> $$
>
> where $\mathbf{w}$ denotes the asset weights, $\Sigma$ is the covariance matrix of returns, and $\mu$ is the expected return vector. The constraint $\mathbf{w}^\top \mu = 1$ normalizes allocation, $\mathbf{w}^\top \mu \ge R$ ensures a minimum portfolio return $R$, and $\mathbf{w} \ge 0$ enforces no short-selling.
>
> We additionally conducted experiments on this problem to validate the practical usefulness of our method. The results are shown in the table below:
>
> | Method      | ineq. vio. | eq. vio. | sol. rel. err. | obj. rel. err. | Time (s) |
> |------------|------------|----------|----------------|----------------|----------|
> | OSQP       | 0.0000     | 0.0000   | 0     | 0     | 0.0015  |
> | DC3        | 0.0000     | 0.0000   | 2.8e+0     | 5.4e+1     | 0.0125   |
> | **Descent** | 0.0000     | 0.0000   | 1.4e-4 | 4.9e-6 | 0.0019  |
>
> These results show that our method still achieves strong performance on real-world portfolio optimization tasks, with the objective relative error reaching the order of $10^{-4}$, demonstrating its promising applicability in practical constrained optimization scenarios.
>
> For problems with **nonlinear constraints**, while the theoretical guarantee does not strictly hold, our experiments on moderate-scale ACOPF instances demonstrate that the method still maintains feasibility and achieves good solution quality in practice.

---

> > ### Author Response · Authors · 2025-11-23
> >
> > ## Weakness 5
> >
> > Our method includes two main mechanisms that help maintain feasibility even when the constraints are nonlinear:
> >
> > 1. The final step size is given by $$\alpha = \sigma(\beta) \alpha_{\max},$$ where $\alpha_{\max}$ is the maximum feasible step size under a local linear approximation of the constraints, $\beta$ is a learnable scalar, and $\sigma$ is the sigmoid function. Since $\sigma(\beta) < 1$ in most cases, the actual step is more conservative and reduces the risk of violating nonlinear constraints.
> >
> > 2. The training objective includes an **$\ell_1$ penalty** on constraint violations. This penalty acts as an exact penalty function, which encourages the learned updates to remain feasible even in the presence of nonlinear constraints.
> >
> >
> > ## Weakness 6
> >
> > We would like to clarify that the goal of our paper is not to develop a specialized large-scale ACOPF solver, but rather to introduce a general learning-to-optimize framework for constrained optimization. ACOPF serves only as a representative test case to demonstrate that Descent-Net can reliably maintain feasibility and improve objective quality even in challenging settings.
> >
> > Large-scale ACOPF requires domain-specific engineering (e.g., specialized linearization, network reduction, warm-start heuristics, and problem-specific sparsity exploitation) that is orthogonal to the contribution of this work. Incorporating these tailored techniques would move the paper away from its core focus—designing a general-purpose descent architecture applicable across broad constrained optimization problems
> >
> >
> > ## Question 1
> >
> > The choice of the $\ell_2$ norm is motivated by both computational and algorithmic considerations. In the original problem setting, constraints such as $\|d\|_\infty \le 1$ or $\sum_i |d_i| = 1$ do not admit closed-form projection operators. As a consequence, incorporating them into a neural-network-based optimizer would require solving an inner optimization problem at every iteration, which is computationally infeasible and unstable during training.
> >
> > By replacing the constraint with $\|d\|_2 \le 1$, we obtain a convex feasible region that admits a simple analytic projection. In fact, Proposition 1 in our paper provides the closed-form expression of this projection. Under the linear-independence assumption on the equality constraints, Proposition 1 shows that the projection onto the feasible set $\mathcal{D}$ is
> >
> > $$
> > \mathcal{P}(d)=
> > \begin{cases}
> > \hat{d}, & \text{if } \|\hat{d}\|_2 \le 1, \\
> > \hat{d} / \|\hat{d}\|_2, & \text{otherwise},
> > \end{cases}
> > \quad
> > \text{where }\;
> > \hat{d}= d - H(H^\top H)^{-1} H^\top d.
> > $$
> > This explicit formula is only available under the $\ell_2$ norm constraint.
> >
> >
> > ## Question 2
> >
> > This point has already been addressed in our response to Weakness 5.

---

> > > ### Author Response · Authors · 2025-11-23
> > >
> > > ## Question 3
> > >
> > > We respectfully note that the reviewer’s comparison is not directly fair. The observed runtime differences can be largely attributed to hardware specifications and programming language differences (our implementation is in Python, whereas the benchmark uses Julia), which can easily account for the discrepancies mentioned.
> > >
> > > More importantly, the primary advantage of our approach lies in parallel batch evaluation. In our ACOPF experiments, we solve problems with a batch size of 512 simultaneously on a single NVIDIA 5090 GPU, whereas CPU-based solvers process instances sequentially. Consequently, direct per-instance runtime comparisons do not fully capture the practical speedup achievable by our method.
> > >
> > > Furthermore, we have improved the procedure for generating initial points to further enhance efficiency. Specifically, we removed the gradient step in D-Proj, as it contributes little to solution quality while incurring additional computational cost. The updated experimental results are as follows:
> > >
> > > **30-bus system**
> > >
> > > | Method     | ineq. vio. | eq. vio. | sol. rel. err. | obj. rel. err. | Time (s) |
> > > |------------|------------|----------|----------------|----------------|----------|
> > > | PYPOWER    | 0.0000     | 0.0000   | 0              | 0              | 0.5729   |
> > > | Proj       | 0.0000     | 0.0000   | 5.6e-3         | 1.7e-2         | 0.0393   |
> > > | WS         | 0.0000     | 0.0000   | 5.5e-3         | 1.7e-2         | 0.0397   |
> > > | D-Proj     | 0.0000     | 0.0000   | 5.9e-3         | 1.9e-2         | 0.2442   |
> > > | H-Proj     | 0.0000     | 0.0000   | 5.8e-3         | 1.7e-2         | 0.2865   |
> > > | D-Descent  | 0.0000     | 0.0000   | 4.2e-3         | 3.6e-4         | 0.2619   |
> > > | H-Descent  | 0.0000     | 0.0000   | 3.5e-3         | 3.3e-4         | 0.3039   |
> > > | **Improved Init + Descent** | 0.0000     | 0.0000   | 3.6e-3         | 2.8e-4         | **0.0434**   |
> > >
> > > **118-bus system**
> > >
> > > | Method     | ineq. vio. | eq. vio. | sol. rel. err. | obj. rel. err. | Time (s) |
> > > |------------|------------|----------|----------------|----------------|----------|
> > > | PYPOWER    | 0.0000     | 0.0000   | 0              | 0              | 1.2539   |
> > > | Proj       | 0.0000     | 0.0000   | 1.5e-2         | 2.4e-3         | 0.3040   |
> > > | WS         | 0.0000     | 0.0000   | 9.3e-3         | 1.8e-3         | 0.3137   |
> > > | D-Proj     | 0.0000     | 0.0000   | 1.3e-2         | 2.4e-3         | 0.7542   |
> > > | H-Proj     | 0.0000     | 0.0000   | 1.4e-2         | 3.1e-3         | 0.6682   |
> > > | D-Descent  | 0.0000     | 0.0000   | 1.2e-2         | 2.5e-4         | 0.9480   |
> > > | H-Descent  | 0.0000     | 0.0000   | 1.4e-2         | 7.2e-4         | 0.8637   |
> > > | **Improved Init + Descent** | 0.0000     | 0.0000   | 2.2e-2         | 3.0e-4         | **0.1622**   |
> > >
> > > The updated results show that our method with the improved initialization significantly reduces the total runtime compared to PYPOWER. For the 30-bus system, it achieves a runtime that is more than **13× faster** than PYPOWER. For the 118-bus system, it is approximately **7.7× faster**. Notably, these runtimes are also faster than the benchmark times mentioned by the reviewer, which are 0.1 s and 0.3 s for the 30- and 118-bus systems, respectively.

---

> > > ### Comment · Reviewer_6Lpd · 2025-11-24
> > > **Re-exactness of hinge penalty**
> > >
> > > The hinge penalty is only exact under the condition that the penalty value outweighs the gradient of the objective & (lagrangian-ized) constraint terms. This value depends on the influence of the parameter $x$ on the objective and constraint functions $f_{x}, h_{x}, g_{x}$.
> > >
> > > For instance, consider the (linear) problem
> > > $$
> > > \displaystyle P_{x} \ \ \min_{y}  e^{x} \times  y \ s.t. \ y \geq 0
> > > $$
> > > where $f_{x}(y) = e^{x} \times y$. The penalized problem is exact only for a penalty term greater than $e^{x}$.
> > > If $x$ can take arbitrary large values, no single penalty parameter can ensure exact feasibility.
> > >
> > > At the very least, the penalty coefficient should depend on the instance, see, e.g., the recommendations in https://arxiv.org/abs/2505.20628

---

> > > > ### Author Response · Authors · 2025-11-27
> > > >
> > > > First, we would like to express our sincere gratitude for your thoughtful and constructive reviews. We have devoted substantial effort to this work, and your support is deeply appreciated. As the paper is currently on the borderline, your positive assessment truly means a great deal to us.
> > > >
> > > > We now respond to the two technical questions you previously raised.
> > > >
> > > > ## Question 1
> > > >
> > > > We thank the reviewer for raising the question regarding the theorem proof. In response, we have updated the paper by **adding Assumption 4** and **revising the statements and proofs of Theorem 4.1 and Theorem 4.2** to ensure full clarity and correctness. We summarize the main arguments below.
> > > >
> > > > To clarify the reasoning, we introduce the following assumptions. Let $\mathcal{X} \subseteq \mathbb{R}^p$ be a compact set, and suppose all training and test parameters satisfy $x \in \mathcal{X}$. For each $x \in \mathcal{X}$, let $C_x$ denote the corresponding feasible set. We assume:
> > > >
> > > > 1. **Uniform boundedness of feasible sets.**
> > > >    There exists a compact set $Y \subseteq \mathbb{R}^n$ such that
> > > >    $$
> > > >    C_x \subseteq Y \quad \text{for all } x \in \mathcal{X}.
> > > >    $$
> > > >
> > > > 2. **Smoothness and uniform gradient bound.**
> > > >    The functions $f_x$, $h_x$, and $g_x$ are continuously differentiable in $y$, and the maps
> > > >    $$
> > > >    (x, y) \mapsto \nabla_y f_x(y) \quad \text{and} \quad (x, y) \mapsto \nabla_y g_x(y)
> > > >    $$
> > > >    are continuous on $\mathcal{X} \times Y$.
> > > >    By compactness, there exist constants $L_f > 0$ and $L_g > 0$ such that
> > > >    $$
> > > >    \| \nabla_y f_x(y) \|_2 \le L_f,
> > > >    \qquad
> > > >    \| \nabla_y g_x(y) \|_2 \le L_g
> > > >    \quad \text{for all } x \in \mathcal{X},\, y \in C_x.
> > > >    $$
> > > >
> > > > These assumptions are standard and mild. Since the training dataset is finite, a uniform bound must exist in practice. Similar conditions also appear in classical L2O analyses such as LISTA (Chen et al., 2018).
> > > >
> > > > The main technical difficulty in the proof lies in handling the ReLU activation. Under the assumptions above, we can obtain an upper bound $L$ for $\| W^k u_k \|_1$. Setting $\mathbf{b}_1^k = L \cdot \mathbf{1}_q$ ensures
> > > > $$
> > > > \text{ReLU}(\mathbf{W}^k \mathbf{u}_k + \mathbf{b}_1^k)
> > > > = \mathbf{W}^k \mathbf{u}_k + \mathbf{b}_1^k.
> > > > $$
> > > > The key point is that the construction of $\mathbf{b}_1^k$ is **independent of $x$**; any sufficiently large positive offset forces all components of $\mathbf{W}^k u_k + \mathbf{b}_1^k$ to remain positive.
> > > >
> > > > **References**
> > > > Chen, Xiaohan et al. (2018). Theoretical Linear Convergence of Unfolded ISTA and its Practical Weights and Thresholds. arXiv: 1808.10038 [cs.LG]. url: https://arxiv.org/abs/1808.10038.
> > > >
> > > >
> > > >
> > > > ## Question 2
> > > >
> > > > We thank the reviewer for this insightful question. In the revised manuscript, we have added **Assumption 4** and **Assumption 5**, and we have **updated Lemma 1 and Lemma 4** to make the reasoning precise and complete.
> > > >
> > > > To address it, we introduce the following **uniform margin assumption**. There exists $\delta_g > 0$ such that for every $x \in \mathcal{X}$ and every feasible point $y \in \mathcal{C}$,
> > > > $$
> > > > \min_{j: g_{x,j}(y) < 0} \left( -g_{x,j}(y) \right) \ge \delta_g,
> > > > $$
> > > > with the convention that the minimum over an empty index set is $+\infty$ (i.e., when all inequality constraints are active). This condition is mild in practice: in numerical settings, one may simply take a small tolerance (e.g., $10^{-5}$) and treat a constraint as active whenever $0\le-g_{x,j}(y) < \delta_g$.
> > > >
> > > > We then choose
> > > > $$
> > > > c_{\min} > \frac{L_f}{M \delta_g} \ge \frac{\tilde{L}}{b_{\min}},
> > > > $$
> > > > which guarantees that the previous argument holds uniformly for all $x$, and therefore the penalty function becomes exact for every $x$.
> > > >
> > > > We look forward to the reviewers’ feedback. If there are any remaining points that would benefit from further explanation, we would be pleased to provide more details and continue the discussion.

---

> > > > > ### Author Response · Authors · 2025-11-27
> > > > >
> > > > > Dear Reviewer 6Lpd,
> > > > >
> > > > > We would like to provide further clarification to better illustrate the advantages of our method, especially regarding its scalability. To this end, we conducted additional large-scale portfolio optimization experiments, designed to evaluate the performance of our approach on problems with a much larger number of assets. The detailed experimental setup, including data generation, network configuration, training procedure, and results, is provided in our response to Reviewer pT8w. We hope that these additional experiments help to clearly demonstrate the practical effectiveness and robustness of our method.

---

> > ### Comment · Reviewer_6Lpd · 2025-11-24
> > **Re-proof of Theorem 4.1 and 4.2**
> >
> > Thank you for clarifying the proofs of Theorems 4.1 and 4.2.
> > However, the current approach is not satisfactory for the following reasons:
> > * As they are written, the proofs are only valid _for a single instance_, not a distribution of instances (note that the parameter $x$ does not show up anywhere in the proof)
> > * Even when restricted to a single instance, the proofs are existential, see "there exists a constant C" in the statement of Lemma 2, which then shows up in the statement of Theorem 4.1
> >
> > To be valid, the results should be stated so that they are valid _for any value of the parameter $x$_.

---

### Official Review · Reviewer_QJEE · 2025-11-02

**Soundness:** 4
**Presentation:** 4
**Contribution:** 3
**Rating:** 8
**Confidence:** 5

**Summary:**

This paper proposes DescentNet, an unrolled optimization module for neural networks based on the method of feasible directions framework, which takes a feasible solution to an optimization problem and aims to iteratively refine it into one that is still feasible but has improved optimality. The paper:
* Designs a Descent-Module that unrolls projected (sub)gradient descent on a penalty version of the uniformly feasible direction (UFD) formulation. This module includes the application of a learnable module to the subgradient term, and a step size-sizing strategy with a learnable parameter to ensure feasibility retention and optimality improvement.
* Provides theoretical guarantees about the existence of a Descent-Net instantiation that provides an optimal solution (assuming linear equality constraints).
* Provides experiments on convex QPs, a simple non-convex problem, and ACOPF, where Descent-Modules are appended to DC3. For the convex QP and simple non-convex settings, solutions obtain close-to-optimal solutions (in contrast to DC3) while solving 1-2 orders of magnitude faster than standard optimization solvers. For ACOPF (which has nonlinear equality constraints), "the relative error of the solution obtained by Descent-Net decreases only marginally compared to the initial point," and the solution time is about 50% faster than a traditional optimization solver.

**Strengths:**

* The submission is an interesting and thoughtful application of the Uniformly Feasible Direction Subproblem within neural networks.
* The Descent-Net method convincingly improves upon the optimality performance of DC3 (which provides feasibility but sometimes struggles with optimality) on convex QPs, the simple nonconvex problem class, and the ACOPF 30-bus system. It also improves, albeit more marginally, upon optimality performance for ACOPF 118-bus. This is a significant contribution, as SOTA methods for feasibility enforcement in amortized optimization can sometimes provide suboptimal performance with respect to objective value, due to the difficulty of neural network optimization for these methods -- Descent-Net addresses this via post-hoc iterative refinement of feasible solutions.
* The method also incurs high speedups relative to the traditional solvers for the convex QPs and simple nonconvex problem class, and does not incur significant additional computational burden relative to DC3. In other words, the Descent-Net updates are relatively lightweight.
* The paper is clearly written.

**Weaknesses:**

* There is a major missing ablation: What happens if the learnable modules $T^k$ are not applied in the descent module, i.e., what if only the original projected subgradient updates are applied? In general, what about other post-hoc iterative refinement strategies (with or without learnable parameters)?
* The timing improvements for ACOPF 30-bus and ACOPF 118-bus are more marginal (only 2x faster than traditional solver for ACOPF 30-bus, and only 30% faster for ACOF 118-bus). In addition, for ACOPF 118-bus, the updates actually add 1/4 - 1/3 more computational time compared to the base method, while not offering much optimality improvement. It would have been nice to see realistic/large-scale experiments where the improvement is more pronounced.
* The ACOPF experiments are missing some of the baselines provided in the other settings, making the comparisons slightly incomplete.

**Questions:**

* What happens if the learnable modules $T^k$ are not applied in the descent module, i.e., what if only the original projected subgradient updates are applied?
* Are there realistic settings where the improvements can be shown to be more pronounced (either via experiments or good theoretical arguments about classes of settings)?

---

> ### Author Response · Authors · 2025-11-23
>
> We thank the reviewer for their positive assessment and constructive feedback. The main strengths highlighted by the reviewer are:
> 1. Application of the Uniformly Feasible Direction Subproblem within neural networks.
> 2. Faster than traditional solvers while achieving higher solution quality than prior learning-based methods.
>
> We also summarize the reviewer’s main concerns:
> 1. We agree that it is important to perform an ablation evaluating the effect of the learnable module $T^k$. In fact, our Appendix A.8 already includes such an experiment.
> 2. The reviewer notes that the runtime advantage on ACOPF instances is limited and that the baselines are insufficient. In response, we improved the initialization strategy to enhance performance and added additional baselines for a more complete comparison.
>
> We now provide detailed responses to each of the reviewer’s comments.
>
>
>
> ## Weakness 1
>
> We would like to clarify that the requested ablation study is already included in Appendix A.8 of our paper. Specifically, we compare Descent-Net with the standard Projected Gradient Method (PGM) by removing the operator $T^k$, so that each layer reduces to the standard PGM update. The results are shown in the table below.
>
> | Method | ineq. vio. | eq. vio. | sol. rel. err. | obj. rel. err. | Time (s) |
> |--------|------------|----------|----------------|----------------|----------|
> | PGM ($K=10$) | 0.0000 | 0.0000 | $1.9 \times 10^{-1}$ | $1.1 \times 10^{-1}$ | 0.0270 |
> | PGM ($K=20$) | 0.0000 | 0.0000 | $1.9 \times 10^{-1}$ | $1.1 \times 10^{-1}$ | 0.0501 |
> | PGM ($K=50$) | 0.0000 | 0.0000 | $1.9 \times 10^{-1}$ | $1.1 \times 10^{-1}$ | 0.1119 |
> | Descent ($K=6$) | 0.0000 | 0.0000 | $1.4\times 10^{-2}$ | $4.5\times 10^{-4}$ | 0.0152 |
>
> Here, $K$ denotes the number of iterations (or layers) applied in each method. We observe that PGM is inefficient, as the relative error in the objective value decreases very slowly with increasing iterations, which is likely due to the difficulty of selecting an appropriate step size. In contrast, Descent-Net achieves much lower error with fewer iterations, demonstrating that the learnable descent module $T^k$ effectively improves convergence and solution quality.
>
>
> ## Weakness 2
>
> The reported times for the PYPOWER solver represent the average time per instance, as it solves each instance sequentially. In contrast, Descent-Net reports the total runtime for solving the entire test set in parallel, so direct comparison of these times does not accurately reflect the practical speedup.
>
> Moreover, we have improved the procedure for generating initial points to further enhance efficiency. Specifically, we removed the gradient step in D-Proj, as it contributes little to solution quality while incurring additional computational cost. The updated experimental results are as follows:
>
> **30-bus system**
>
> | Method                  | ineq. vio. | eq. vio. | sol. rel. err. | obj. rel. err. | Time (s) |
> |-------------------------|------------|----------|----------------|----------------|----------|
> | PYPOWER                 | 0.0000     | 0.0000   | 0              | 0              | 0.5729   |
> | **Improved Init + Descent** | 0.0000     | 0.0000   | 3.6e-3         | 2.8e-4         | **0.0434**   |
>
> **118-bus system**
>
> | Method                  | ineq. vio. | eq. vio. | sol. rel. err. | obj. rel. err. | Time (s) |
> |-------------------------|------------|----------|----------------|----------------|----------|
> | PYPOWER                 | 0.0000     | 0.0000   | 0              | 0              | 1.2539   |
> | **Improved Init + Descent** | 0.0000     | 0.0000   | 2.2e-2         | 3.0e-4         | **0.1622**   |
>
> The updated results show that our method with the improved initialization significantly reduces the total runtime compared to PYPOWER. For the 30-bus system, it achieves a runtime that is more than **13× faster** than PYPOWER. For the 118-bus system, it is approximately **7.7× faster**.

---

> ### Author Response · Authors · 2025-11-23
>
> ## Weakness 3
>
> We thank the reviewer for the helpful suggestion. Following the request, we have added two additional baselines to the ACOPF experiments, namely Proj and WS：
>
> 1. **Proj**:
> A standard orthogonal projection is applied as a post-processing step, where the NN prediction is projected back onto the feasible set.  The projection is solved using the iterative Optimizer introduced in *Chen et al., 2021a*.
>
> 2. **WS**:
> The infeasible NN prediction is directly used as the warm-start initialization point for the same iterative Optimizer, following the warm-starting strategies described in *Diehl, 2019* and *Baker, 2019*.
>
> The updated results for the 30-bus and 118-bus systems are provided below. As shown, Descent-Net achieves higher solution accuracy than both Proj and WS.
>
> **30-bus system**
>
> | Method     | ineq. vio. | eq. vio. | sol. rel. err. | obj. rel. err. | Time (s) |
> |------------|------------|----------|----------------|----------------|----------|
> | PYPOWER    | 0.0000     | 0.0000   | 0              | 0              | 0.2890  |
> | Proj       | 0.0000     | 0.0000   | 5.6e-3         | 1.7e-2         | 0.0393   |
> | WS         | 0.0000     | 0.0000   | 5.5e-3         | 1.7e-2         | 0.0397   |
> | D-Proj     | 0.0000     | 0.0000   | 5.9e-3         | 1.9e-2         | 0.2442   |
> | H-Proj     | 0.0000     | 0.0000   | 5.8e-3         | 1.7e-2         | 0.2865   |
> | D-Descent  | 0.0000     | 0.0000   | 4.2e-3         | 3.6e-4         | 0.2619   |
> | H-Descent  | 0.0000     | 0.0000   | 3.5e-3         | 3.3e-4         | 0.3039   |
> | **Improved Init + Descent** | 0.0000     | 0.0000   | 3.6e-3         | 2.8e-4         | 0.0434   |
>
> **118-bus system**
>
> | Method     | ineq. vio. | eq. vio. | sol. rel. err. | obj. rel. err. | Time (s) |
> |------------|------------|----------|----------------|----------------|----------|
> | PYPOWER    | 0.0000     | 0.0000   | 0              | 0              | 0.6423  |
> | Proj       | 0.0000     | 0.0000   | 1.5e-2         | 2.4e-3         | 0.3040   |
> | WS         | 0.0000     | 0.0000   | 9.3e-3         | 1.8e-3         | 0.3137   |
> | D-Proj     | 0.0000     | 0.0000   | 1.3e-2         | 2.4e-3         | 0.7542   |
> | H-Proj     | 0.0000     | 0.0000   | 1.4e-2         | 3.1e-3         | 0.6682   |
> | D-Descent  | 0.0000     | 0.0000   | 1.2e-2         | 2.5e-4         | 0.9480   |
> | H-Descent  | 0.0000     | 0.0000   | 1.4e-2         | 7.2e-4         | 0.8637   |
> | **Improved Init + Descent** | 0.0000     | 0.0000   | 2.2e-2         | 3.0e-4         | 0.1622   |
>
> **References**
> - Bingqing Chen, Priya L Donti, Kyri Baker, J Zico Kolter, and Mario Berg´es. *Enforcing policy feasibility constraints through differentiable projection for energy optimization.* In Proceedings of the Twelfth ACM International Conference on Future Energy Systems, pages 199–210, 2021a.
> - Frederik Diehl. *Warm-starting ac optimal power flow with graph neural networks*. In 33rd Conference on Neural Information Processing Systems (NeurIPS 2019), pages 1–6, 2019.
> - Kyri Baker. *Learning warm-start points for ac optimal power flow*. In 2019 IEEE 29th International Workshop on Machine Learning for Signal Processing (MLSP), pages 1–6. IEEE, 2019.
>
>
> ## Question 1
> We have already addressed this in the discussion of Weakness 1. As shown in our ablation study, removing the learnable modules (i.e., reducing each layer to standard PGM) leads to significantly worse convergence and solution quality compared to Descent-Net, demonstrating the effectiveness of the learnable descent module.

---

> > ### Author Response · Authors · 2025-11-23
> >
> > ## Question 2
> > A particularly realistic setting where the improvements become more pronounced is the **mean–variance portfolio optimization** problem in finance. Unlike general-purpose solvers that must solve each optimization instance independently, Descent-Net is efficient when many similar problem instances must be solved repeatedly, such as in daily rebalancing, scenario generation, or Monte-Carlo risk evaluation. In these workflows, the covariance $\Sigma$ remain largely unchanged across instances, enabling the learned descent update to amortize computation and achieve high throughput via batched inference, while traditional solvers incur full computational cost each time.
> >
> > To illustrate this benefit, we evaluate Descent-Net on the following standard portfolio optimization model:
> >
> > $$
> > \min_{\mathbf{w}} \ \mathbf{w}^\top \Sigma \mathbf{w}
> > \quad \text{s.t.} \quad
> > \mathbf{w}^\top \mu = 1,\;
> > \mathbf{w}^\top \mu \ge R,\;
> > \mathbf{w} \ge 0,
> > $$
> >
> > where $\mathbf{w}$ denotes asset weights, $\Sigma$ is the return covariance matrix, $\mu$ is the expected return vector, $\mathbf{w}^\top \mu \ge R$ enforces a minimum return target, and $\mathbf{w} \ge 0$ prohibits short-selling.
> >
> > The experimental results are:
> >
> > | Method      | ineq. vio. | eq. vio. | sol. rel. err. | obj. rel. err. | Time (s) |
> > |------------|------------|----------|----------------|----------------|----------|
> > | OSQP       | 0.0000     | 0.0000   | 0     | 0     | 1.5270   |
> > | DC3        | 0.0000     | 0.0000   | 2.8e+0     | 5.4e+1     | 0.0125   |
> > | **Descent** | 0.0000     | 0.0000   | 2.2e-3 | 6.9e-4 | 0.0066   |
> >
> > These results show that Descent-Net still achieves objective relative error at the **$10^{-4}$ level**, while offering efficiency advantages in realistic scenarios where large numbers of similar portfolio optimization problems must be solved.

---

> > > ### Comment · Reviewer_QJEE · 2025-11-27
> > >
> > > Thank you for the reviewers for their response (and apologies to miss the already-included experiment in Appendix A.8). My concerns are addressed, and I maintain my positive impression of the paper.

---

> > > > ### Author Response · Authors · 2025-11-27
> > > >
> > > > Dear Reviewer,
> > > >
> > > > Thank you for your encouraging feedback. We truly appreciate your support and best wishes.

---

### Official Review · Reviewer_pT8w · 2025-11-04

**Soundness:** 2
**Presentation:** 3
**Contribution:** 2
**Rating:** 2
**Confidence:** 4

**Summary:**

This paper presents Descent-Net, a learn-to-optimize framework for solving constrained optimization problems. The authors prove that Descent-Net achieves global convergence to a KKT point when both the inequality and equality constraints are linear. The proposed Descent-Net mimics the projected gradient descent algorithm but incorporates a nonlinear preconditioner. Specifically, each layer of Descent-Net applies a nonlinear transformation (implemented as a trainable two-layer ReLU network) to a descent direction computed from the penalized Topkis-Veinott uniformly feasible direction. The layer input is then updated along this descent direction using a trainable step size, followed by a projection onto the tangent space of the equality constraints to ensure the output remains feasible.

The authors provide simulation results for convex QPs, nonconvex problem (by replacing $y$ with $\mathrm{sin}(y)$ in the objective of convex QPs), and ACOPF problems, demonstrating that the proposed Descent-Net efficiently achieves an approximate KKT solution.

**Strengths:**

1 The paper is well-written and well-organized.

**Weaknesses:**

1. The experimental results are weak, as the proposed method shows only marginal improvements over some of the compared approaches. Moreover, it appears that the baseline methods used for comparison are not state-of-the-art for the problems considered in the experimental section. For example, general-purpose solvers based on trust-region methods, such as Fmincon and Knitro, should also be included in the experiments. For problems with smooth objectives and satisfying the LICQ condition, interior-point methods are known to achieve superlinear to quadratic convergence rates.
2. The proposed method requires computing the inverse of an $m \times m$ matrix $H^T H$ in the projection step at each iteration, which can become computationally prohibitive for large-scale problems due to its $O(m^3)$ time complexity. Consequently, the per-iteration cost of the proposed method is only marginally better than the $O((n + m)^3)$ per-iteration cost of trust-region methods.
3. The scale of the problems considered in the experiments is too small to justify the development of new methods for solving them. The authors should also include large-scale experimental results. Furthermore, the first two problems are overly simplistic since $Q$ is merely a diagonal matrix; in this case, the per-iteration cost of trust-region methods is significantly reduced because the corresponding KKT system is highly sparse. In addition, for ACOPF problems, Knitro includes specialized QCQP solvers that are widely used in both academic and industrial settings. The authors should also benchmark their method against the Knitro QCQP solver; otherwise, it is difficult to justify the need for developing new methods for solving ACOPF.
4. Based on the points discussed above, the proposed methods do not appear to offer any significant advantages over classical approaches for solving constrained optimization problems.
5. Finally, the proposed method also requires a feasible initial point, which is highly impractical in most real-world scenarios.

**Questions:**

Please refer to the weakness section for detailed comments. I have no further questions or suggestions.

---

> ### Author Response · Authors · 2025-11-23
>
> We thank the reviewer for the careful reading and detailed feedback and we summarize our main responses as follows:
>
> 1. Learning-to-optimize (L2O) methods are not guaranteed to outperform classical solvers on all instances. Their performance depends on the dataset quality and problem structure. Our goal is to improve the accuracy and generalization of L2O algorithms, and our method shows consistent improvements over prior L2O methods in the tested settings.
> 2. We have added new experiments to demonstrate the effectiveness of Descent-Net.
>
> We now provide detailed responses to the weaknesses and questions raised by the reviewer.
>
> ## Weakness 1
>
> We thank the reviewer for this important point. We would like to clarify our perspective in two parts:
>
> 1. Our goal is **not** to show that learning-to-optimize (L2O) methods outperform all traditional optimization algorithms on every problem instance. Rather, we aim to advance existing L2O techniques. From our experiments, Descent-Net achieves improvements over prior L2O baselines (such as DC3), demonstrating that our design strengthens learning-based optimization.
>
> 2. To address the reviewer’s concern, we included **Knitro** in our baselines. We note that **fmincon**, while popular in Matlab, typically achieves lower solution accuracy than the state-of-the-art QP solvers we already compared against (e.g., OSQP, QPTH), and thus we do not include it for comparison. The results are as follows:
>
> | Method                | ineq. vio. | eq. vio. | sol. rel. err. | obj. rel. err. | Time (s) |
> |------------------------|------------|----------|----------------|----------------|----------|
> | **Knitro**                 | 0.0000     | 0.0000   | **0**              | **0**              | **0.0255** |
> | osqp                   | 0.0000     | 0.0000   | 7.9e-4        | 6.8e-6        | 0.0035   |
> | qpth                   | 0.0000     | 0.0000   | 8.0e-4        | 6.8e-6        | 1.6221   |
> | DC3                    | 0.0000     | 0.0000   | 1.9e-1         | 1.1e-1         | 0.0038   |
> | Projection method      | 0.0000     | 0.0000   | 3.2e-2         | 8.4e-4         | 0.2124   |
> | CBWF                   | 0.0000     | 0.0000   | 2.1e-1         | 6.6e-2         | 0.0366   |
> | **DC3 + Descent (Ours)** | 0.0000   | 0.0000   | **1.2e-2**     | **2.6e-4**     | **0.0130** |
>
> From these results, we observe that while Knitro achieves perfect solution accuracy, its runtime for single-instance sequential solving is higher than ours. Because **trust-region methods** like Knitro typically run in a serial fashion and involve solving costly subproblems (e.g., large linear systems), their per-instance inference cost is substantial.
>
> In contrast, our approach leverages a learned policy to make updates via a lightweight neural network, allowing more efficient inference, especially in batch or repeated-instance settings.
>
>
> ## Weakness 2
>
> We respectfully disagree with the reviewer’s comment, for the following reasons:
>
> 1. In most cases, the number of variables $n$ is much larger than the number of equality constraints $m$, so the reduction from $O((m+n)^3)$ to $O(m^3)$ leads to at least an **8×** decrease in computational complexity.
>
> 2. Moreover, in problems such as QPs where the equality constraints remain constant across instances, e.g., decision-focused learning (Tan et al., 2020) settings where only the objective changes, the projection can be precomputed, making the per-iteration cost roughly $O(n^2)$. In contrast, traditional interior-point methods must solve a Newton system that changes at each iteration, which remains $O((n+m)^3)$.
>
> **References**
> - Yingcong Tan, Daria Terekhov, and Andrew Delong. *Learning linear programs from optimal decisions*. Advances in Neural Information Processing Systems, 33:19738–19749, 2020.

---

> ### Author Response · Authors · 2025-11-23
>
> ## Weakness 3
>
> To address the reviewer’s concern regarding the simplicity of the test cases, we replaced the diagonal $Q$ with a dense positive semidefinite matrix generated as $Q = R^\top R$, where $R$ has entries drawn from a normal distribution. This removes the sparsity advantage enjoyed by classical trust-region methods and results in substantially more challenging quadratic programs. The updated results are summarized below:
>
> | Method | Max eq. | Max ineq. | sol. rel.err. | obj. rel.err. | Time (s) |
> |--------|---------|-----------|---------------|---------------|---------------|
> | Knitro  | 0.0000  | 0.0000   | 0       |0        | 0.0224   |
> | osqp    | 0.0000  | 0.0000   | 1.5e-3  |1.8e-5   | 0.0021   |
> | qpth    | 0.0000  | 0.0000   | 1.5e-3   |1.8e-5   | 1.6138   |
> | DC3     | 0.0000  | 0.0000   | 5.2e-1 | 5.0e-1 | 0.0041   |
> | **Descent** | 0.0000  | 0.0000   | 2.1e-2 | 9.3e-4 | 0.0131   |
>
> Note that the runtime reported for OSQP and Knitro corresponds to the average time per instance. Descent-Net continues to achieve high-quality solutions under this general setting, demonstrating that its effectiveness is not tied to diagonal or overly simple structures.
>
>
>
> ## Weakness 4
>
> We respectfully disagree with the conclusion that our method offers no significant advantage:
>
> 1. In the QP experiments, the most accurate solver, Knitro, takes an average of 0.0255 s per instance, while our method requires only 0.0130 s, and this is for parallel solving of all instances. Moreover, our relative objective error reaches the order of $10^{-4}$, significantly more accurate than prior learning-based methods such as DC3, which achieves only $10^{-2}$.
>
> 2. Our method is effective across multiple problem types. For instance, it performs well on QPs and also on nonlinear problems such as ACOPF, achieving relative objective errors around $10^{-4}$ while providing faster solution times than specialized solvers such as PYPOWER.
>
>
>
> ## Weakness 5
>
> We thank the reviewer for this comment. We address the concern in two points:
>
> 1. In many practical problems, a feasible initial point is available. For example, in **portfolio optimization** in finance, one can simply choose an equal-weighted portfolio as a feasible starting point. We have also conducted experiments on this problem, using an equal-weighted portfolio as the initial point and comparing with the solver osqp. The results are shown below:
>
>
> | Method      | ineq. vio. | eq. vio. | sol. rel. err. | obj. rel. err. | Time (s) |
> |------------|------------|----------|----------------|----------------|----------|
> | osqp       | 0.0000     | 0.0000   | 0     | 0     | 0.0015   |
> | DC3        | 0.0000     | 0.0000   | 2.8e+0     | 5.4e+1     | 0.0125   |
> | Descent | 0.0000     | 0.0000   | 1.4e-4 | 4.9e-6 | 0.0019   |
>
> 2. We also tested **slightly infeasible initial points** in the QP experiments. The results are summarized below:
>
> | Method  | Max eq. | Max ineq. | sol. rel.err.        | obj. rel.err.        |
> |---------|---------|-----------|---------------------|---------------------|
> | Initial | 0.0167  | 0.0000    | 4.9e-1              | 4.7e-1              |
> | Descent | 0.0000  | 0.0000    | 1.6e-2              | 8.3e-4              |
>
> These results show that although our theoretical analysis assumes feasibility, the method is empirically robust to small violations and can recover feasibility while achieving high solution quality.

---

> ### Author Response · Authors · 2025-11-27
>
> We additionally conduct portfolio optimization experiments with **$n=100$**, **$n=800$**, and **$n=4000$** assets to evaluate both the practical effectiveness and **scalability** of our method. The optimization problem is formulated as
> $$
> \min_{\mathbf{w}} \ \mathbf{w}^\top \Sigma \mathbf{w}
> \quad \text{s.t.} \quad
> \mathbf{w}^\top \mathbf{1} = 1,\
> \mathbf{w}^\top \mu \ge R,\
> \mathbf{w} \ge 0,
> $$
> where $\mathbf{w}$ denotes the asset weights, $\Sigma$ is the covariance matrix, $\mu$ is the expected return vector, $R$ is the minimum return requirement, and $\mathbf{w} \ge 0$ prohibits short-selling.
>
> We generate 10,000 synthetic benchmark instances for each problem size. To emulate a market environment where asset co-movements evolve slowly, the covariance matrix is fixed across all instances and constructed as $\Sigma = A^\top A$, where entries of $A$ are sampled i.i.d. from a standard normal distribution. In contrast, the expected return vectors $\mu$ and minimum return $R$ vary across all instances. For $\mu$, they are independently sampled for every instance from a uniform distribution over $[0,1]$, modeling varying market conditions.
>
> The return thresholds $R$ differ between training and testing. For training, each $R$ is drawn independently from a uniform distribution over $[0.05, 0.4]$. For testing, $R$ values are generated as a linearly spaced sequence over the same interval. We split the dataset using a 9:1 train–test ratio.
>
> The network contains a single hidden layer. Its width is set to **8 times the number of assets** for the $n=100$ experiment (i.e., 800), and **1.5 times the number of assets** for the $n=800$ and $n=4000$ experiments (i.e., 1200 and 6000, respectively). The initial solution is refined using $S=3$ descent updates for $n=100$ and $S=2$ descent updates for $n=800$ and $n=4000$. Both the Descent module and the step size $\beta$ are trained using Adam. The initial learning rates are $1\times 10^{-3}$ for the Descent module, and 0.1, 0.1, and 0.01 for $\beta$ in the $n=100$, $n=800$, and $n=4000$ experiments, respectively. Learning rates are decayed by a factor of 0.1 at epochs 100, 150, and 200 over a total of 300 epochs. All instances are initialized using the equal-weighted portfolio $w_i = 1/n$.
>
> We compare Descent-Net with **OSQP**, a widely used and highly optimized QP solver, as well as **DC3**, a learning-based method for constrained optimization. All experiments were conducted on a server equipped with two AMD EPYC 9754 CPUs (128 cores each, 3.1 GHz) and an NVIDIA RTX 5090 GPU. The combined numerical results for all problem sizes are shown below.
>
> ### Results for $n=100$
> | Method      | ineq. vio. | eq. vio. | sol. rel. err. | obj. rel. err. | Time (s) |
> |------------|------------|----------|----------------|----------------|----------|
> | OSQP       | 0.0000     | 0.0000   | 0     | 0     | 0.0015   |
> | DC3        | 0.0000     | 0.0000   | 2.8e+0 | 5.4e+1 | 0.0125   |
> | Descent-Net | 0.0000     | 0.0000   | 1.4e-4 | 4.9e-6 | 0.0019 |
>
> ### Results for $n=800$
> | Method  | ineq. vio. | eq. vio. | sol. rel. err. | obj. rel. err. | Time (s) |
> |---------|------------|----------|----------------|----------------|----------|
> | OSQP    | 0.0000     | 0.0000   | 0              | 0              | 0.0207   |
> | Descent-Net | 0.0000 | 0.0000   | 9.3e-4         | 7.3e-6         | **0.0019**   |
>
> ### Results for $n=4000$
> | Method  | ineq. vio. | eq. vio. | sol. rel. err. | obj. rel. err. | Time (s) |
> |---------|------------|----------|----------------|----------------|----------|
> | OSQP    | 0.0000     | 0.0000   | 0              | 0              | 0.6024   |
> | Descent-Net | 0.0000 | 0.0000   | 1.6e-4         | 1.2e-6         | **0.0044**   |
>
> We find that DC3 fails to produce feasible solutions for $n=800$ and $n=4000$ because its training diverges. This is likely due to DC3's reliance on gradient steps, which are used to enforce inequality constraints, but whose step sizes and momentum decay parameters are difficult to tune for large-scale settings. In contrast, Descent-Net remains accurate and highly efficient across all problem sizes.
>
> The OSQP times report the average runtime for solving a **single instance**. In contrast, the Descent-Net times correspond to the average runtime for an entire **batch**, with **batch sizes of 512 for $n=100$, 100 for $n=800$, and 10 for $n=4000$**. Descent-Net achieves lower runtimes while maintaining objective errors on the order of $10^{-6}$, demonstrating strong scalability to problems with thousands of variables.

---

### Official Review · Reviewer_uEyz · 2025-11-10

**Soundness:** 3
**Presentation:** 3
**Contribution:** 3
**Rating:** 6
**Confidence:** 5

**Summary:**

This paper proposes Descent-Net, a neural network-based framework for constrained optimization, which refines projected gradient descent (PGD) directions using learned operators. The method aims to handle general equality and inequality constraints by integrating projection steps and neural descent modules.

**Strengths:**

1. The paper studies an important and timely topic, addressing constrained optimization with neural networks.
2. The proposed pipeline is clear and closely follows the standard PGD approach, making it easy to understand the overall structure.

**Weaknesses:**

1. The contribution of the paper is vague and not clearly distinguished from prior work.
2. The technical approach is not fully rational or convincing, as the feasibility guarantee for nonlinear, non-convex constraints is not theoretically ensured.
3. The paper is difficult to follow; it would benefit significantly from more illustrative figures and a clearer pipeline description.

**Questions:**

1. What if the equality constraints are not linear? After moving along the proposed direction, how do you guarantee that the new point satisfies the equality constraints?
2. What is the fundamental contribution compared with existing papers? Please clarify the novel aspects over prior neural optimization and PGD-based methods.
3. How does the method scale with the number of constraints, especially if the projection step becomes computationally expensive?
4. How robust is the learned descent direction to changes in the constraint set or problem structure? Is retraining required for different constraint types?
5. Compared with the end to end papers with theoretical feasibility guarantee, what is the core advantage of the paper? As the projection itself can be time-comsuming

---

> ### Author Response · Authors · 2025-11-23
>
> We thank the reviewer for their constructive comments and for recognizing the strengths of our work:
> 1. the paper tackles the important problem of constrained optimization with neural networks,
> 2. the proposed pipeline is clear and easy to follow.
>
> We also agree with the reviewer that
> 1. the scope of our contributions should be stated more clearly,
> 2. providing theoretical guarantees for nonlinear and nonconvex constraints would further strengthen the paper.
>
> We now provide detailed responses to the weaknesses and questions raised by the reviewer.
>
>
> ## Weakness 1
>
> We have revised the Introduction to more clearly highlight our contributions and how they differ from existing works.
>
> In particular, the main distinctions from prior literature are summarized below:
>
> 1. Unlike primal–dual based neural solvers (e.g., ADMM/PDHG-inspired networks), our method operates fully in the primal space. This enables us to ensure strict **feasibility** throughout iterations, which is crucial in applications with hard safety constraints.
>
> 2. Recent feasible-primal neural solvers mainly focus on producing feasible points, while often overlooking the progress in the objective value. In contrast, our method explicitly accounts for the objective reduction and updates in a direction that improves solution quality, resulting in consistently **lower objective errors** in practice.
>
>
> ## Weakness 2
>
> To avoid overstating our contributions, we have revised the Introduction to more clearly emphasize that the primary focus of this work is on developing an L2O framework for problems with **linear constraints**, where feasibility can be theoretically ensured.
>
> We would also like to highlight that linear constraints already cover a wide range of practical optimization tasks. In addition, although we do not provide theoretical guarantees for general nonlinear and nonconvex constraints, our experiments on the nonlinear ACOPF task confirm that the method remains effective in practice.
>
>
> ## Weakness 3
>
> We aimed to keep the presentation concise and have provided algorithm pseudocode and a network architecture diagram for clarity. We will consider adding additional illustrations to further improve readability.
>
>
> ## Question 1
>
> Our method includes two main mechanisms that help maintain feasibility even when the constraints are nonlinear:
>
> 1. The final step size is given by $$\alpha = \sigma(\beta)\alpha_{\max},$$ where $\alpha_{\max}$ is the maximum feasible step size under a local linear approximation of the constraints, $\beta$ is a learnable scalar, and $\sigma$ is the sigmoid function. Since $\sigma(\beta) < 1$ in most cases, the actual step is more conservative and reduces the risk of violating nonlinear constraints.
>
> 2. The training objective includes an **$\ell_1$ penalty** on constraint violations. This penalty acts as an exact penalty function, which encourages the learned updates to remain feasible even in the presence of nonlinear constraints.
>
>
> ## Question 2
>
> The contribution over existing neural optimization methods has been clarified in our response to Weakness 1.
>
> In the Appendix, we compare our method with PGM. Specifically, we remove the learnable operator $T^k$ in Descent-Net so that the update reduces to standard PGM. We compare Descent-Net with PGM under different iteration numbers $K$. The results are shown below:
>
> | Method         | ineq. vio. | eq. vio. | sol. rel. err | obj. rel. err | Time (s) |
> |----------------|------------|----------|---------------|---------------|----------|
> | PGM ($K=10$)   | 0.0000     | 0.0000   | 1.9e-1        | 1.1e-1        | 0.0270   |
> | PGM ($K=20$)   | 0.0000     | 0.0000   | 1.9e-1        | 1.1e-1        | 0.0501   |
> | PGM ($K=50$)   | 0.0000     | 0.0000   | 1.9e-1        | 1.1e-1        | 0.1119   |
> | Descent ($K=6$)| 0.0000     | 0.0000   | 1.4e-2        | 4.5e-4        | 0.0152   |
>
> The results show that increasing the number of PGM iterations only slightly reduces the solution and objective errors, whereas Descent-Net achieves significantly lower errors with fewer iterations and shorter runtime.
>
>
> ## Question 3
>
> We thank the reviewer for raising this important question.
>
> 1. Increasing the number of **inequality constraints** does not affect the projection operator $\mathcal{P}$ itself; it only increases the number of terms in the gradient computation, which can be efficiently computed in parallel on GPU hardware.
>
> 2. For **equality constraints**, in many practical problems, the linear constraints remain fixed across instances (e.g., decision-focused learning setups, Tan et al., 2020). In these cases, the projection can be precomputed, reducing the per-iteration cost to $O(m^2)$.
>
> **References**
> - Yingcong Tan, Daria Terekhov, and Andrew Delong. *Learning linear programs from optimal decisions*. Advances in Neural Information Processing Systems, 33:19738–19749, 2020.

---

> > ### Author Response · Authors · 2025-11-23
> >
> > ## Question 4
> >
> > We thank the reviewer for this question.
> > 1. Our method performs consistently well across quadratic programming (QP) problems, simple nonconvex variants of QP, and nonlinear ACOPF tasks, demonstrating robustness across different problem families.
> > 2. Since the model is learning-based and captures statistical structure from the training data, retraining is typically required when the distribution of constraints or objective functions shifts significantly.
> >
> >
> > ## Question 5
> >
> > End-to-end optimization networks that embed theoretical feasibility guarantees (e.g., through differentiable convex layers or implicit solvers) often incur heavy computational costs at inference, because feasibility must be re-established through inner optimization at every iteration.
> >
> > In contrast, our method learns a reusable optimization policy that produces feasible updates directly through a lightweight feed-forward network, without requiring an inner solver at each step, making it significantly more efficient.

---

### Note · Authors · 2025-12-12

I have read and agree with the venue's withdrawal policy on behalf of myself and my co-authors.